# Cap analogs with a hydrophobic photocleavable tag enable facile purification of fully capped mRNA with various cap structures

Masahito Inagaki[1,8], Naoko Abe [ID][1,8], Zhenmin Li[1], Yuko Nakashima[1,2], Susit Acharyya[1], Kazuya Ogawa[1], Daisuke Kawaguchi[1], Haruka Hiraoka [ID][1], Ayaka Banno[1], Zheyu Meng[1], Mizuki Tada[1], Tatsuma Ishida[1], Pingxue Lyu[1], Kengo Kokubo[1], Hirotaka Murase [ID][1], Fumitaka Hashiya[2], Yasuaki Kimura[1], Satoshi Uchida [ID][3,4,5] & Hiroshi Abe [ID][1,2,6,7] [✉]

Starting with the clinical application of two vaccines in 2020, mRNA therapeutics are currently being investigated for a variety of applications. Removing immunogenic uncapped mRNA from transcribed mRNA is critical in mRNA research and clinical applications. Commonly used capping methods provide maximum capping efficiency of around 80–90% for widely used Cap-0- and Cap-1-type mRNAs. However, uncapped and capped mRNA possesses almost identical physicochemical properties, posing challenges to their physical separation. In this work, we develop hydrophobic photocaged tag-modified cap analogs, which separate capped mRNA from uncapped mRNA by reversed-phase high-performance liquid chromatography. Subsequent photo-irradiation recovers footprint-free native capped mRNA. This approach provides 100% capping efficiency even in Cap-2-type mRNA with versatility applicable to 650 nt and 4,247 nt mRNA. We find that the Cap-2-type mRNA shows up to 3- to 4-fold higher translation activity in cultured cells and animals than the Cap-1-type mRNA prepared by the standard capping method.

Decades of research into mRNA therapeutics[1,2] have culminated in two approved mRNA vaccines for the COVID-19 pandemic in 2020[3,4]. The 5′ cap plays a critical role in mRNA therapeutics, wherein $N^7$-methylguanosine ($m^7G$) is attached to the 5′ end of mRNA via a 5′–5′ triphosphate bridge (Fig. 1a)[5,6]. The interaction between the 5′ cap and the poly-A chain at the 3′ end of mRNA enhances the translation activity of mRNA[5,6]. The other roles of the cap structure include splicing, protection to nuclease, and nuclear transport[5,6]. Structurally, the 2′-hydroxyl groups at the 5′ end of mRNA are methylated in the cap structure[5–8]. The cap structures of the first two nucleotides possessing zero, one, and two 2′ O- methyl (OMe) groups are called Cap-0, Cap-1, and Cap-2, respectively (Fig. 1a). The methylation in the cap structure is

[1]Department of Chemistry, Graduate School of Science, Nagoya University, Furo-cho, Chikusa-ku, Nagoya, Aichi 464-8602, Japan. [2]Research Center for Materials Science, Nagoya University, Furo-cho, Chikusa-ku, Nagoya, Aichi 464-8602, Japan. [3]Graduate School of Medical Science, Kyoto Prefectural University of Medicine, 1-5 Shimogamohangi-cho, Sakyo-ku, Kyoto 606-0823, Japan. [4]Innovation Center of NanoMedicine (iCONM), Kawasaki Institute of Industrial Promotion, 3-25-14 Tonomachi, Kawasaki-ku, Kawasaki 210-0821, Japan. [5]Medical Research Institute, Tokyo Medical and Dental University (TMDU), 1-5-45 Yushima, Bunkyo-ku, Tokyo 113-8510, Japan. [6]CREST, Japan Science and Technology Agency, 7, Gobancho, Chiyoda-ku, Tokyo 102-0076, Japan. [7]Institute for Glyco-core Research (iGCORE), Nagoya University, Furo-cho, Chikusa-ku, Nagoya, Aichi 464-8601, Japan. [8]These authors contributed equally: Masahito Inagaki, Naoko Abe. [✉]e-mail: h-abe@chem.nagoya-u.ac.jp

a marker of "self" mRNA, making mRNA with less immunogenic[9–11]. A recent study provides insight into the identity and functions of Cap-2 cap structure in mRNA, which have remained unknown for years[12]. Cap-2 structure drastically reduces the mRNA affinity to retinoic acid-inducible gene-I (RIG-1), an innate immune receptor recognition compared to Cap-1 structure, with moderately increasing mRNA stability and translation activity[12]. In mammals, when the first transcribed base of capped mRNA is A, most of its $N^6$ position is methylated, forming $m^6A$[5,6]. The presence of $m^6A$ increases resistance to an mRNA-decapping enzyme, Dcp2[13].

In chemical or semi-chemical methods of capped RNA synthesis[14–17], the length of the RNA strand is limited to about 150 bases. For synthesizing longer capped mRNA with high efficiency, RNA polymerase of phage origin is mainly used, especially T7 RNA polymerase (T7 RNAP). The current capping methods include enzymatic and co-transcriptional capping[5]. In the enzymatic capping, transcribed mRNA is treated with a capping enzyme[5,18], often followed by methyltransferase treatment to convert the Cap-0 structure to the Cap-1[5]. In co-transcriptional capping, a synthetic cap analog, i.e., $m^7G$-containing nucleotide, is added to the in vitro transcription (IVT) reaction[19,20]. The first generation dinucleotide cap analog $m^7G(5')ppp(5')G$ had the problem of the chain-elongation starting from the 3' hydroxyl group of the $m^7G$. The unintended elongation results in 30–50% of the reversely capped RNA being inactive for translation[20]. This issue was solved by an anti-reverse cap analog [ARCA, $m_2^{7,3'-O}G(5')ppp(5')G$] methylated in the 3'-hydroxyl group of $m^7G$, ensuring the chain-elongation exclusively from the desired side[21,22]. Recently, new cap analogs consisting of trinucleotides and tetranucleotides have been developed[23–27] to introduce the Cap-1 or Cap-2 structure into mRNA directly. Despite such progress, co-transcriptional capping still has a challenge in obtaining capped mRNA with 100% efficiency because nucleotides such as GTP compete with cap analogs for the initiation step of the transcription reaction.

The uncapped byproduct has a triphosphate (ppp) at the 5' end, a motif found in viral mRNA, triggering an immune response via innate immune receptors such as RIG-1 and melanoma differentiation-associated protein 5 (MDA5)[11,28]. Moradian et al. have reported that a small amount of 5' ppp-RNA byproduct from a standard co-transcriptional capping method induces substantial immune responses[29], wherein the capping method provides around 80–90% capping efficiency[30]. These studies highlight the necessity to maximize capping efficiency or minimize uncapped ppp-RNA contaminants. The enzymatic treatment is a standard method to remove uncapped mRNA byproducts. For example, the 5'-terminal triphosphate of uncapped RNA will be converted to the monophosphate by RNA 5' polyphosphatase, followed by degrading monophosphorylated RNA using XRN I, a 5'→3' exoribonuclease requiring 5' monophosphate[24,26,27,31]. However, considering mRNA susceptibility to non-specific degradation, the number of purification processes should be minimized. In addition, repeated enzymatic treatment significantly increases the production cost, hampering commercial mass preparation of mRNA. These issues motivate us to develop a non-enzymatic methodology for diminishing the uncapped mRNA immunogenicity.

Physical separation of capped mRNA from uncapped mRNA would become a simple, robust, cost-effective purification strategy. However, capped and uncapped mRNA have almost the same physicochemical properties, posing challenges for their separation.

In this study, we design hydrophobic cap analogs, namely Pure-Cap analogs, for purifying capped mRNA. PureCap cap analogs are integrated into the mRNA 5'-terminal during the IVT reaction. Its hydrophobicity enables the purification of the capped mRNA using reversed-phase high-performance liquid chromatography (RP-HPLC) (Fig. 1b, c). Using a photodegradable 2-nitrobenzyl derivative as a hydrophobic tag enables tag removal under mild conditions by light irradiation[32]. This method relies on RP-HPLC, a standard purification process for nucleic acid medicine, rather than enzymatic treatment. Especially, purification of mRNA by RP-HPLC is an established method for eliminating double-stranded RNA (dsRNA) contaminants produced during the IVT, which increases the immunogenicity of mRNA and inhibits its translation[33–37]. Thus, the PureCap method can isolate capped mRNA from the two significant impurities, uncapped ppp-RNA and dsRNA, simultaneously providing a practical advantage in mRNA production.

Using the PureCap technology, we succeeded in footprint-free purification of the capped mRNA in a native form with 100% capping efficiency, which to the best of our knowledge, has not been achieved previously for the first time. The PureCap mRNA shows undetectable immunogenicity even without enzymatic treatment. Achieving 100% efficiency in various cap species, our approach provides an unbiased platform to study their structure-activity relationship by excluding the influence of different capping efficiency observed among cap species. Notably, we prepare fully capped Cap-2 mRNA, which shows up to three- to four-fold higher translation activity in cultured cells and animals compared with mRNA prepared by standard capping methods.

## Results

### Design and synthesis of dinucleotide PureCap analogs

We designed and synthesized PureCap cap analogs modified with a hydrophobic tag (Figs. 1c, 2 and Supplementary Figs. 1–14). To begin with, four dinucleotide PureCaps were synthesized. The structure of a hydrophobic tag contains a *tert*-butyl (*t*Bu) group in a 2-nitrobenzyl (Nb) photocaging molecule[32], where *t*Bu moiety works to enhance its hydrophobicity and chemical stability[38]. In three of the four designed cap analogs, the Nb tag was introduced into either the 2' or 3' hydroxyl group of $m^7G$. DiPure (**1**) is an analog that links the Nb group via an acetal group at the 2' position and has a free 3' hydroxyl group since methylation of the 2'-hydroxyl has been reported to inhibit chain elongation from $m^7G$ as well as that of the 3'-hydroxyl[39]. The other two analogs have *O*-methyl (OMe) and *O*-Nb modifications at the 2'/3' positions of $m^7G$. Analogs with OMe modifications on either 2' or 3' were named DiPure/2'OMe (**2**) or DiPure/3'OMe (**3**), respectively. Using the ONb and OMe modifications, all three analogs have anti-reverse activity. Being deprotected after introduction into mRNA, three types of purely capped mRNA can be prepared that differ only in the presence or absence of the OMe modification. As a result, we planned to make a precise comparison in the translation activity with the structure without OMe modification, which was impossible before. The remaining compound, DiPure/N2 (**4**), is a cap analog modified with the Nb at the exocyclic amino group of $m^7G$.

The key step in synthesizing cap analogs is synthesizing the corresponding diphosphate by phosphorylation of guanosine derivatives. For the synthesis of cap analogs known as the ARCA derivatives, protocols consisting of monophosphorylation by the Yoshikawa method using phosphoryl chloride, activation of monophosphate through phosphorimidazolide formation, and diphosphate formation by reaction with alkylammonium phosphate were applied[19]. However, this method is complicated because it is based on the stepwise introduction of phosphate groups. It requires multiple purifications with aqueous solvents, such as ion exchange and reversed-phase chromatography. The introduction of a lipophilic moiety, such as in the PureCap analogs, is expected to decrease the solubility of the intermediates in aqueous solvents, making purification particularly difficult. Therefore, we applied a one-pot synthesis of the diphosphate directly from the nucleoside to avoid the solubility problem and simplify and shorten the synthetic processes. In the first approach, phosphoryl chloride is applied to a guanosine derivative, and the diphosphate is obtained by adding an alkylammonium phosphate salt to the resulting phosphorodichloridate intermediate[40–42]. The second

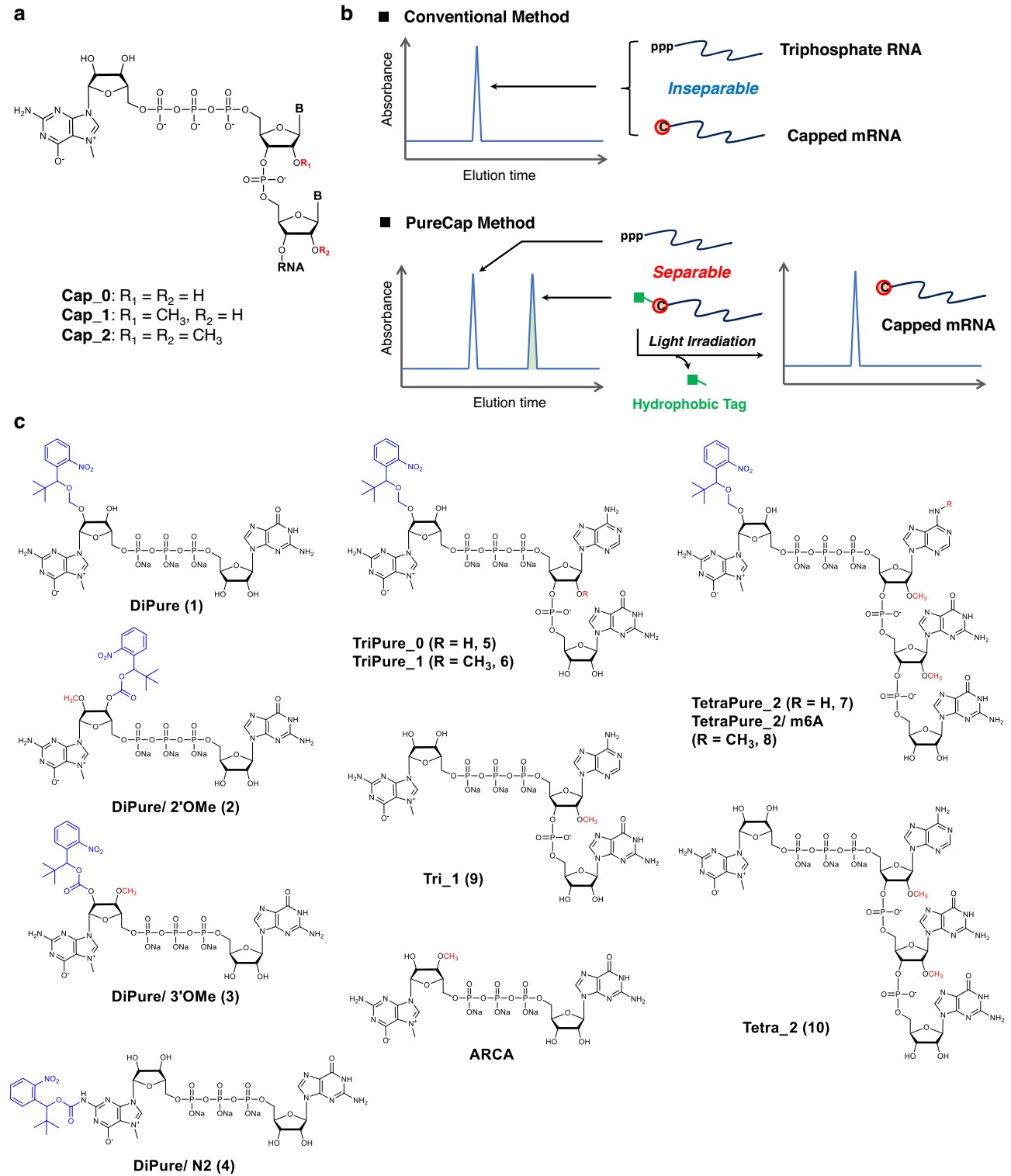

**Fig. 1 | Hydrophobic cap analogs for the purification of capped mRNA using RP-HPLC (the PureCap method). a** Structure of the 5′ cap in eukaryotic mRNA. **b** Capped mRNA can be isolated from uncapped impurities using RP-HPLC by the hydrophobicity of a tag attached to the cap moiety. After the isolation of the capped mRNA, light irradiation can remove the tag. **c** Structure of cap analogs modified with a photodegradable hydrophobic tag, "the PureCaps," and untagged control cap analogs synthesized/evaluated in this study.

approach is to obtain the diphosphate directly from 5′-tosylated guanosine by reacting with tetrabutylammonium pyrophosphate salt[43–46]. The resulting diphosphate is converted to the desired dinucleotide PureCap analog by methylation at the $N^7$ position and subsequent condensation reaction with guanosine monophosphate imidazolide in the presence of zinc chloride.

2′-$O$-Nb-modified guanosine **14** as a phosphorylation precursor was synthesized from $N^2$-isobutyrylguanosine **11** in four steps (Fig. 2a). First, $N^2$-isobutyrylguanosine **11** was reacted with 1,3-dichloro-1,1,3,3-tetraisopropyldisiloxane (TIPDSCl$_2$) in pyridine to afford the 3′,5′-$O$-protected compound in a quantitative yield[47]. The key Nb group was introduced by reaction with nitrobenzyl alcohol methylthioacetal **12** in

**Fig. 2 | Synthesis of PureCap analogs. a** Synthesis of 7-methyl 2′-O-Nb-guanosine 5′ diphosphate imidazolide (**16**). **b** Synthesis of dinucleotide PureCap analog DiPure (**1**). **c** Synthesis of tri/tetranucleotide PureCap analogs; TriPure_0 (**5**), TriPure_1 (**6**), TetraPure_2 (**7**), and TetraPure_2/ m6A (**8**).

the presence of *N*-iodosuccinimide (NIS) and TfOH in THF at −40 °C to afford the corresponding product **13** in 76% yield. The silyl protecting group was subsequently removed with tetrabutylammonium fluoride to afford the unprotected compound in 75% yield, which was then treated with 28% aqueous ammonia at 55 °C to remove the isobutyryl group to afford 2′-O-Nb modified guanosine **14** in 70% yield. In the first approach, **14** was reacted with phosphoryl chloride in trimethyl phosphate, and then tetrabutylammonium monophosphate salt was applied to afford the corresponding diphosphate **15** in one pot after hydrolysis (Supplementary Fig. 3). Although **15** could be synthesized by this method in a short step, the low yield of the diphosphate formation (13%) was a problem. Therefore, direct diphosphorylation via 5′-O-tosylated form **14** was investigated as a second approach. First, 2′-O-Nb guanosine (**14**) was reacted with tosyl chloride in pyridine to obtain the 5′-O-tosylated form in 60% yield, which was then reacted with tetrabutylammonium diphosphate salt to give the diphosphate **15** in 54% yield. The resulting diphosphate was reacted with methyl iodide in DMSO to afford the 7-methylguanosine diphosphate derivative in 72% yield. This compound was then converted to the corresponding phosphorimidazolide **16** by treatment with imidazole, 2,2′-dithiodipyridine, and triphenylphosphine in 89% yield. Finally, compound **17** was reacted with guanosine monophosphate phosphoroimidazolide **18** in the presence of zinc chloride to afford PureCap analog DiPure (**1**) in 50% yield (Fig. 2b). 2′-O-methylated analog DiPure/2′OMe (**2**), 3′-

methylated analog DiPure/3′OMe (**3**), and *N*²-Nb-protected analog DiPure/N2 (**4**) were synthesized similarly (Supplementary Figs. 6–8).

**Design and synthesis of tri/tetranucleotide PureCap analogs**
The cap structure produced by the co-transcription of dinucleotide cap analogs is Cap-0. However, in 2009, Ishikawa et al. reported that mRNAs with the Cap-1 structures could be prepared using trinucleotide cap analogs[23]. Sikorski et al. recently synthesized a series of trinucleotide cap analogs and compared their translation activity of mRNA containing Cap-1 with that of mRNA containing Cap-0, reporting a higher translation activity of the former in some mammalian cultured cells[24]. Methylation of the terminus 3′ hydroxyl of the trinucleotide cap analogs cannot introduce the Cap-2 structure into mRNA, as is known as the rationale for developing ARCA[21,22,39]. To this end, tetranucleotide cap analog are required to introduce a Cap-2 structure in mRNA, as reported recently by ref. 27.

We synthesized tri- or tetranucleotide PureCap analogs in this study to produce quantitatively capped mRNAs with Cap-1 and Cap-2 structures (Figs. 1c, 2c). We selected A and m6A bases as the base of the sites to which m7G links in the tri- or tetranucleotides; according to the report that the highest translation activity with Cap-1 mRNA was observed when the base in the cap structure was A or m6A[24].

Sikorski et al. reported the synthesis of trinucleotide cap analogs by reacting 5′-monophosphorylated dinucleotides synthesized by a

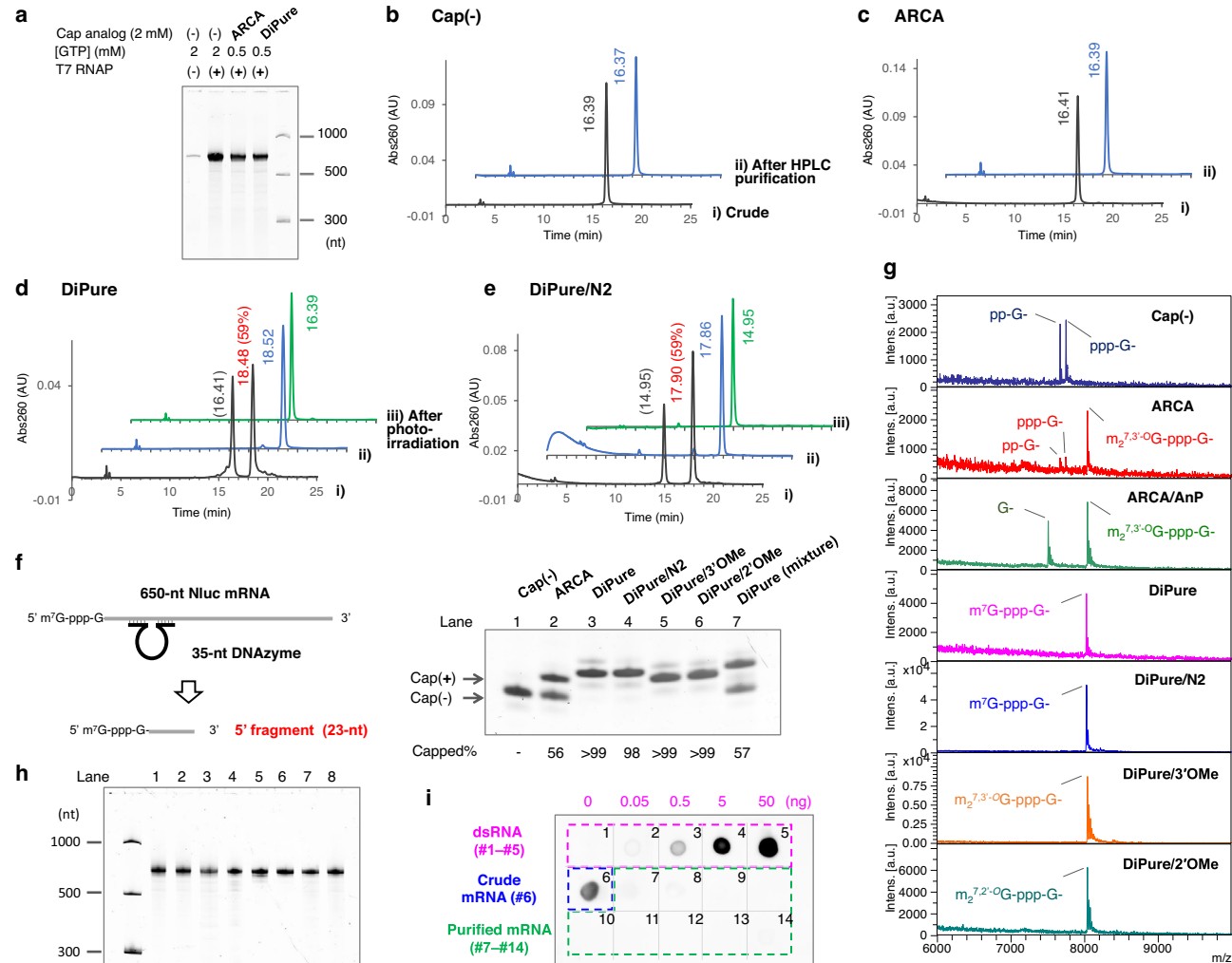

**Fig. 3 | IVT and purification of mRNA by RP-HPLC using dinucleotide PureCap analogs.** Nluc mRNA (650-nt) was transcribed using T7 RNAP in the absence or presence of a cap analog. **a** Representative dPAGE analysis of the transcription reaction. **b–e** RP-HPLC analysis of the RNA transcript. A cap analog added to the reaction was indicated in the figure. The transcript was analyzed as a crude mixture (**i**, black line) or after being purified by preparative HPLC (**ii**, blue line). Purified RNA was analyzed after deprotection by irradiating 365 nm light (**iii**, green line). The elution time (min) of the peak was noted nearby. A ratio (%) of capped mRNA calculated based on the peak area was listed in parentheses after the elution time. The elution time in the experiment shown in **e** differed from others (**b–d**) because HPLC conditions were not completely identical. **f–i** Analysis of purified mRNA used to measure the biological activities. The mRNAs were named after the cap analog used. Cap(−): mRNA prepared without cap analog. AnP: mRNA treated with Antarctic phosphatase. DiPure (mixture): mRNA transcribed in the presence of DiPure

but without isolation of capped mRNA. (**f, g**) Analysis of the 5′ end of purified mRNA cleaved with DNAzyme to assess its capping state. **f** dPAGE analysis of the cleaved 5′ RNA fragments. **g** MALDI-TOF MS analysis of the 5′ RNA fragments. For these observed peaks, the calculated and measured MS agreed well, shown in Supplementary Table 1. **h** dPAGE analysis of purified mRNAs (50 ng each). Lane 1, Cap(−); 2, ARCA; 3, ARCA/AnP; 4, DiPure; 5, DiPure/N2; 6, DiPure/3′OMe; 7, DiPure/2′OMe; 8, DiPure (mixture). (**i**) Removal of dsRNA from purified mRNA (250 ng each) was confirmed by dot-blot assay using anti-dsRNA J2 antibody. As a positive control of the assay, the dsRNA sample was spotted in spots 1 to 5 (0, 0.050, 0.50, 5.0, and 50 ng, respectively). Analyzed mRNAs were as follows; spot 6, mRNA before HPLC purification prepared with no cap analog; 7, Cap(−); 8, ARCA; 9, ARCA/AnP; 10, DiPure; 11, DiPure/N2; 12, DiPure/3′OMe; 13, DiPure/2′OMe; 14, DiPure (mixture). **a–g** Each experiment was repeated independently at least three times to obtain similar results. Source Data are provided with this paper.

solid phase method with $N^7$-methylguanosine diphosphate imidazolide in the presence of zinc chloride[24]. Senthilvelan et al. also reported the synthesis of trinucleotide cap analogs using 5′-monophosphorylated dinucleotides synthesized by a liquid phase method[26]. The 5′-monophosphorylated dinucleotides (**19, 20**) and 5′-monophosphorylated trinucleotides (**21, 22**) synthesized by these synthetic methods were reacted with $N^7$-methyl-2′-O-Nb-guanosine diphosphate imidazolide **16** in the presence of zinc chloride to synthesize PureCap analogs in trinucleotide (**5, 6**) and tetranucleotide (**7, 8**) forms. (Fig. 2c).

## Preparation and purification of mRNA using dinucleotide PureCap analogs

First, we examined whether the synthesized dinucleotide cap analog DiPure could isolate and purify capped mRNA by RP-HPLC as planned.

Using 676-bp double-stranded DNA containing the T7 promoter and encoding NanoLuc luciferase (Nluc) as a template[48], mRNA of 650 bases was transcriptionally synthesized using T7 RNAP. DiPure was added to this reaction, and the resulting RNA transcripts were analyzed by RP-HPLC (Fig. 3). Control samples were prepared without the cap analog and with the commercially available ARCA [$m_2^{7,3'-O}$G(5′)ppp(5′)G]. First, as is common in capped mRNA preparation reactions using ARCA to increase the introduction of the cap analog, the GTP concentration was reduced to 0.5 mM, and NTPs other than GTP and cap analog concentrations were set to 2 mM for the transcription reaction. (Fig. 3a). The resulting RNA transcripts were analyzed by RP-HPLC, and RNA was eluted at 16.4 min in case no cap analog was added or with ARCA (Fig. 3b, c). On the other hand, with the cap analog DiPure, a peak at 18.5 min was observed in addition to the peak at 16.4 min (Fig. 3d).

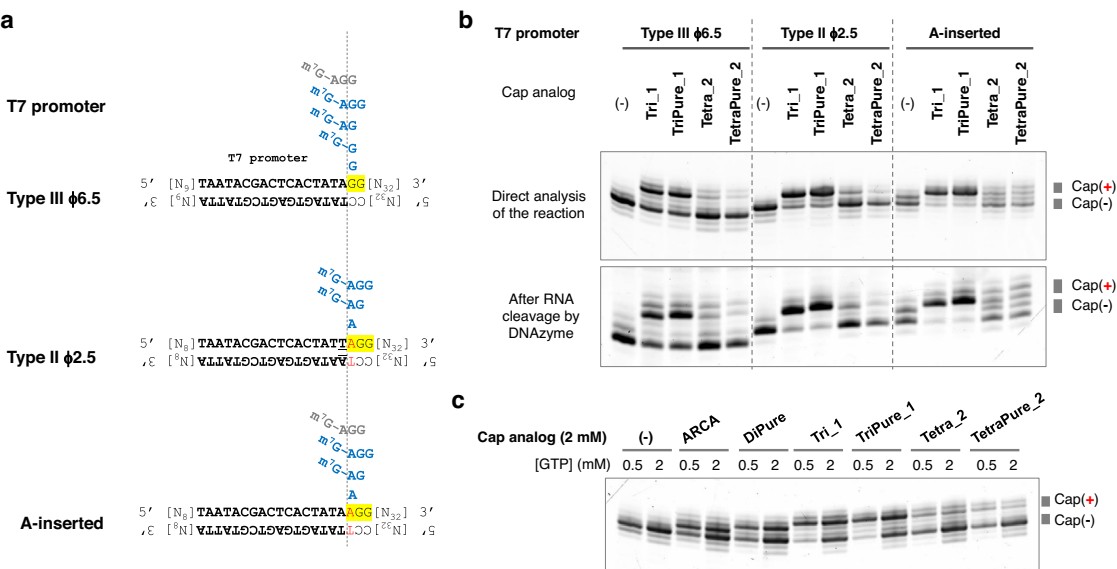

**Fig. 4 | Tri/tetranucleotide PureCap analogs are incorporated by T7 RNAP in a model system. a** Three types of promoter sequences were tested using 60-bp dsDNA. The estimated position of cap analogs when they were incorporated into RNA was depicted in the figure. Major or minor species were shown in blue or gray color, respectively. **b, c** dPAGE analysis of the reaction. An aliquot was taken from the reaction mixture and analyzed (upper panel in **b, c**). Or they were analyzed after being cleaved by DNAzyme 10–23 to align the heterogeneous 3′ ends of the RNA

transcripts (lower panel in **b**). **b** Effect of the promoter sequence on the incorporation of the cap analogs. The concentrations of NTPs and cap analogs were 2 mM. **c** Analysis of IVT on the template with Type III ϕ6.5 promoter. NTP other than GTP and cap analog concentrations were set at 2 mM, and GTP concentrations were set at either 0.5 or 2 mM. Each experiment was repeated independently at least three times to obtain similar results. Source Data are provided with this paper.

We judged that the elution time of capped RNA containing DiPure changed from 16.4 to 18.5 min due to its hydrophobicity (Fig. 3d). To confirm this, the peak at 18.5 min was collected, and 365-nm light was irradiated to the RNA. The elution time of the RNA changed from 18.5 to 16.4 min due to the removal of the hydrophobic protecting group. To confirm the structure of the 5′ end of the RNA, 650-nt Nluc mRNA purified by RP-HPLC was cleaved at 23 bases from the 5′ end by DNAzyme 10–23[49]. The product was subjected to denaturing polyacrylamide gel electrophoresis (dPAGE) and mass analysis (Fig. 3f, g and Supplementary Table 1). The results showed that the mRNA prepared with ARCA had a capping rate of 57% and was a mixture of capped and uncapped mRNA, whereas the mRNA purified by RP-HPLC with DiPure contained no uncapped mRNA. For all mRNAs, including uncapped mRNA, minor bands were observed above and below the main band of the cleavage product, which we believe is due to the 5′ end heterogeneity of the transcript produced by T7 RNAP (Fig. 3f)[50]. These results indicate that capped mRNA can be quantitatively prepared by RP-HPLC purification using the DiPure cap analog.

DiPure/N2, a cap analog with a protecting group at the $N^2$ position of m7G, was similarly added to the IVT reaction, and the generated RNA was analyzed by RP-HPLC. As a result, it was found that capped mRNA could be isolated and purified for mRNA containing this analog using the hydrophobicity of the cap analog (Fig. 3e).

When T7 RNAP transcribes RNA, it is known that highly immunogenic dsRNA is produced as a byproduct[33–37]. When this dsRNA is included in the mRNA as an impurity, the translation activity of the mRNA is significantly impaired due to the triggering of a cellular immune response[24,33,35]. Purification by RP-HPLC or removal using cellulose powder is known for removing dsRNA impurity[24,33,35]. In our preliminary study, we also noticed that purifying capped RNA by RP-HPLC increased the translation activity in HeLa cells by about 50–100-fold. Detection experiments using anti-dsRNA J2 antibody showed that this increase in activity was due to the removal of dsRNA by RP-HPLC purification. Using RNA transcripts with no cap analog added, we examined the elution time of the dsRNA impurity by RP-HPLC and found that it eluted about 2–3 min later than the main product, 650-nt

mRNA, under our analytical conditions (Supplementary Fig. 15). This position coincides with the elution position of mRNA containing PureCap analogs. Therefore, mRNA containing PureCap had to be purified once by HPLC preparative purification of capped mRNA using its hydrophobicity, followed by deprotection by light irradiation. Then a second HPLC purification was added for complete dsRNA removal. Control mRNA samples were also purified by HPLC twice to match the number of cycles in the HPLC purification process.

Capped Nluc mRNAs using DiPure/3′OMe and DiPure/2′OMe were also prepared (Supplementary Fig. 16). In this case, we examined the transcription reaction conditions. We found that the capped mRNA yield after RP-HPLC isolation increased if the GTP concentration was not reduced in the reaction and kept at 2 mM, the same as other NTPs, although the incorporation rate of the PureCap analog decreased. Therefore, in subsequent experiments, the GTP concentration was not reduced, and the desired pure-capped mRNA was isolated from the resulting RNA transcript mixture by RP-HPLC.

The purity of the resulting capped mRNAs was confirmed to be nearly identical by dPAGE (Fig. 3h). A detection assay for dsRNA contamination using the anti-dsRNA antibody confirmed that dsRNA was similarly removed from the capped mRNAs (Fig. 3i). These highly purified capped mRNAs were used for subsequent immunogenicity and translation activity evaluations.

**Preparation and purification of mRNA using tri/tetranucleotide PureCap analogs**

First, a trinucleotide PureCap analog TriPure_1 and a tetranucleotide PureCap analog TetraPure_2 were tested to determine whether they could be incorporated in the IVT using a short RNA transcription system on 60-bp dsDNA templates (Fig. 4). As a control experiment, the efficiency of incorporation of untagged cap analogs was evaluated to know the effect of the modification by comparison. We tested three different promoter sequences: the first was a sequence called Type III ϕ6.5, known as the most versatile and powerful promoter sequence; the second was Type II ϕ2.5, initiated from ATP, anticipating the previously reported 5′ end high uniformity[51,52]. And third, a modified T7

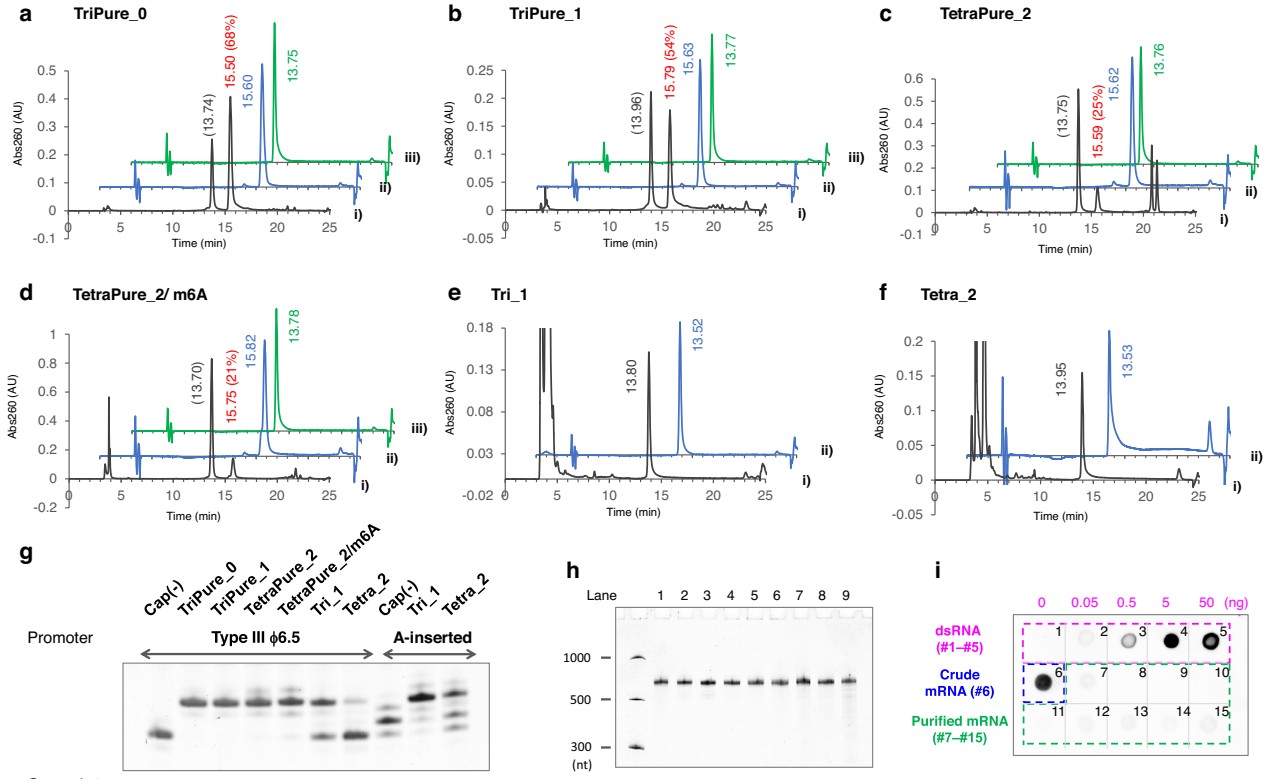

**Fig. 5 | Nluc mRNA (650-nt) preparation using tri/tetranucleotide PureCap analogs. a–f** RP-HPLC analysis of IVT-RNAs. PureCap analogs TriPure_0 (**a**), Tri-Pure_1 (**b**), TetraPure_2 (**c**), and TeraPure_2/m6A (**d**) were added to the reaction. Control mRNA using cap analogs Tri_1 (**e**) or Tetra_2 (**f**) was transcribed from a DNA template containing the A-inserted promoter. The transcript was analyzed as a crude mixture (**i**, black line) or after being purified by preparative HPLC (**ii**, blue line). Purified RNA was analyzed after deprotection by irradiating 365-nm light (**iii**, green line). The elution time (min) of the peak was noted nearby. A ratio (%) of capped mRNA calculated based on the peak area was listed in parentheses after the elution time. **g–i** Analysis of purified mRNA used to measure the biological activities. The mRNAs were named after the cap analog used. Cap(−) means no cap analog was added. AnP means it was further treated with Antarctic phosphatase. **g** dPAGE analysis of the 5′ end of purified mRNA cleaved with DNAzyme 10–23 to

assess its capping state. The capped ratio was calculated based on the band intensities of the capped and uncapped RNA fragments. **h** dPAGE analysis of purified mRNAs (25 ng each). Lane 1, Cap(−); 2, TriPure_0; 3, TriPure_1; 4, TetraPure_2; 5, TetraPure_2/m6A; 6, Tri_1; 7, Tri_1/AnP; 8, Tetra_2; 9, Tetra_2/AnP. **h** Removal of dsRNA from purified mRNA was confirmed by dot-blot assay using anti-dsRNA J2 antibody. 250 ng mRNAs were spotted. As a positive control of the assay, the dsRNA sample was dotted in spots 1 to 5 (0, 0.050, 0.50, 5.0, and 50 ng, respectively). Analyzed mRNAs were as follows; spot 6, mRNA before HPLC purification prepared with no cap analog; 7, Cap(−); 8, TriPure_0; 9, TriPure_1; 10, TetraPure_2; 11, TetraPure_2/m6A; 12, Tri_1; 13, Tri_1/AnP; 14, Tetra_2; 15, Tetra_2/AnP. **a–i** Each experiment was repeated independently at least three times to obtain similar results. Source Data were provided with this paper.

promoter with a corresponding base A-inserted at the +1 position of the ϕ6.5 promoter, referred to as the "A-inserted" promoter in the figure, was tested that is reported to improve the trinucleotide cap analog incorporation in IVT[25]. First, IVT reactions were performed with all NTPs and cap analog at 2 mM concentration, and transcripts were analyzed by dPAGE (Fig. 4). To resolve the 3′ end heterogeneity of the purified RNA transcripts and to facilitate confirmation of the product, the RNA was digested by DNAzyme at ten bases upstream from the 3′ end of the RNA[49]. They were analyzed by dPAGE in the same manner. The results showed that the incorporation efficiency of the trinucleotide cap analog was higher than that of the tetranucleotide for all promoters used and that tag introduction to the trinucleotide did not affect the incorporation efficiency. Indeed, with the A-inserted promoter, the generation of the uncapped transcript was lower, and the selectivity of the transcription initiation from the trinucleotide cap was higher[25]. Unlike the trinucleotide counterpart, a reduction in the incorporation efficiency due to the introduction of the tag was observed for the tetranucleotide cap analogs. Next, we used the Type III ϕ6.5 promoter to determine whether decreasing the GTP concentration increases the amount of capped RNA obtained (Fig. 4c)[27]. The results showed that lowering the concentration of GTP, which competes with the cap analog in the initiation phase, increased the capping ratio but decreased the amount of capped RNA produced.

When using the PureCap analog, the amount of uncapped RNA was not an issue because RP-HPLC could separate the capped RNA. From these results, we chose to keep the GTP concentration the same as the other NTPs in the transcription reaction with PureCap analogs to prepare the mRNA. As for the lower incorporation efficiency of the tetranucleotide cap analogs regardless of the promoter type, it was speculated that its steric bulk makes it difficult to stabilize the initiation complex formed between the tetranucleotide, template DNA, and polymerase[53].

Nluc mRNA was then prepared using these PureCap analogs to study the effects of Cap-0, Cap-1, and Cap-2 structures on translation activity (Fig. 5). Cap-0 and Cap-1 structures were introduced into mRNA using the corresponding trinucleotide PureCaps, TriPure_0, TriPure_1, respectively (Fig. 1a, c). Cap-2 structures were introduced using the tetranucleotide PureCaps TetraPure_2 or TetraPure_2/m6A. Preliminary experiments were performed to compare the amount of capped RNA produced by the two promoters shown in Fig. 4, Type III ϕ6.5 and A-inserted. As a result, we found that the Type III ϕ6.5 promoter made the highest amount of the capped mRNA and that it was appropriate to use this promoter to prepare mRNA (Supplementary Fig. 17). Nluc mRNA was transcribed with 2 mM concentrations of NTPs and cap analog from DNA containing the Type III ϕ6.5 promoter. As shown in Fig. 5, mRNAs with PureCap analogs showed different elution times than uncapped mRNAs, and RP-HPLC could isolate capped

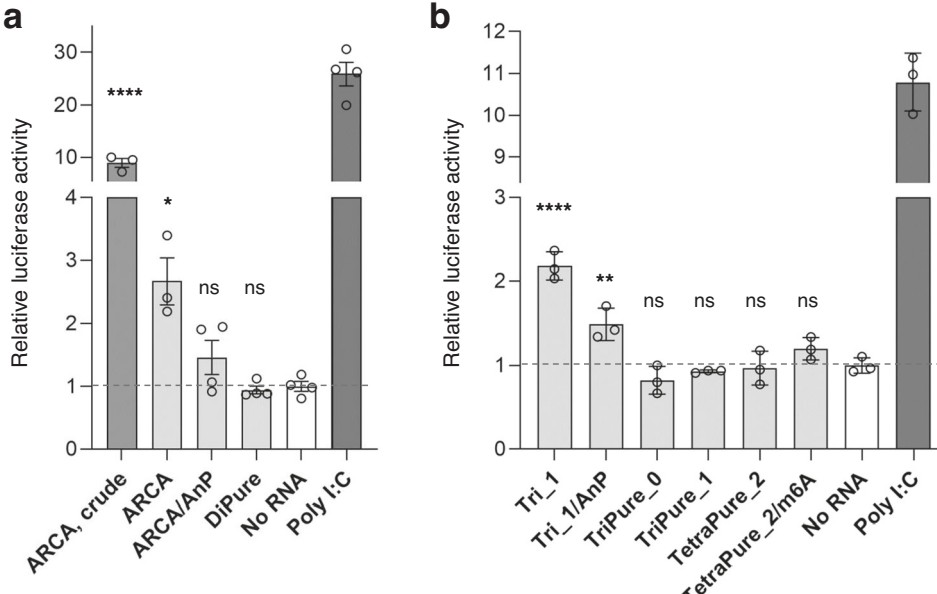

**Fig. 6 | Evaluation of the immunostimulatory effect of capped mRNAs using NF-κB reporter (Luc)–HEK293 cells.** 650-nt Nluc mRNA prepared/ purified using dinucleotide cap analogs (**a**) or tri/tetranucleotide cap analogs (**b**) was transfected into the cells using Lipofectamine MessengerMAX reagent. Poly I: C was used as the positive control of this assay. The cap analog used for mRNA preparation is indicated in the graph. AnP means the mRNA was further treated with Antarctic phosphatase to remove its 5′ phosphate. ARCA-capped RNA before the HPLC purification was also analyzed (ARCA, crude). The expression level was normalized to the mock-transfected control sample (No RNA). Data were mean ± s. e. m. (standard error of the mean) for biological replicates ($n = 3$ or 4). The statistically significant differences of Nluc mRNA-transfected samples from the mock-transfected sample in the one-way ANOVA test followed by the Dunnett test were marked as follows: ns, $p > 0.05$; *, $p < 0.05$; **, $p < 0.01$; ***, $p < 0.001$. Each experiment was repeated independently at least twice to obtain similar results. Source Data are provided with this paper.

mRNAs. Consistent with the previous results producing short RNA transcripts, the incorporation efficiency of tetranucleotide PureCaps was lower than those of trinucleotide PureCaps: 68 and 54% for TriPure_0 and TriPure_1, and 25 and 21% for TetaPure_2, TetraPure_2 /m6A, respectively. To check the identity of the 5′ end of the HPLC-purified mRNAs, they were cleaved using DNAzyme 10–23 at 23 bases from the 5′ end and analyzed by dPAGE and MALDI-TOF MS (Fig. 5g and Supplementary Fig. 18). The capped mRNA produced using PureCap analogs showed almost no band derived from uncapped RNA and observed molecular weight of the 5′ end fragments were consistent with theoretical ones. Control mRNAs were prepared using two cap analogs without tag, a trinucleotide cap analog Tri_1 and a tetranucleotide cap analog Tetra_2 from templates containing two different promoters, Type III φ6.5 and A-inserted one, and the differences in their capping efficiencies were examined (Fig. 5g and Supplementary Fig. 18). As shown in Fig. 5g, both cap analogs were incorporated into mRNA at higher ratios when the template containing A-inserted promoter was used[25]. Therefore, the control mRNAs were transcribed from the template containing the A-inserted promoter and utilized for subsequent biological activity measurements.

### Immunogenicity of mRNAs prepared using PureCap analogs
Mammalian cells have an immune system as a defense mechanism against infection by foreign organisms such as viruses[11,28]. Externally introduced RNA is recognized by pattern recognition receptors such as Toll-like receptor 3 (TLR3), TLR7, TLR8, and RIG-I, triggering an immune response in host cells[28]. If the prepared mRNA contains uncapped 5′ ppp-RNA and/or dsRNA impurity, the defense mechanism against antiviral infection will be activated[11,28]. When they are sensed by RIG-1, a cellular-type immune response occurs, activating protein kinase R and inhibiting intracellular translation. 5′ ppp-RNA is also recognized by the antiviral proteins Interferon-induced protein with tetratricopeptide repeats 1 (IFIT1) and IFIT5[54]. Therefore, removing as much uncapped mRNA as possible when preparing capped mRNAs is essential, which may cause unwanted immune responses.

Since highly purified, quantitatively capped 650-nt Nluc mRNAs were prepared by the PureCap method described above, the immunostimulatory properties of these mRNAs were evaluated using NF-κB reporter (Luc)-HEK293 cell line[55]. Control mRNAs and mRNAs prepared by the PureCap method were introduced into the cells by lipofection, cultured, and lysed. The activation level of the NF-κB pathway was measured as the expression of the firefly luciferase reporter gene. In this system, after activating proinflammatory cytokines or agonists of lymphokine receptors, endogenous NF-κB transcription factors bind to the DNA response elements, inducing the luciferase reporter gene transcription. As shown in Fig. 6, the expression level of the luciferase gene by mRNAs prepared with the PureCap analogs was almost the same as that of the mock-transfected sample (No RNA). In contrast, when mRNA was prepared using the control cap analogs, ARCA or Tri_1 was transfected, and 2.7- and 2.2-fold higher luciferase expression was observed than in the mock-transfected sample. The expression decreased when the mRNA was dephosphorylated using Antarctic phosphatase, which indicates that their immunogenicity was attributed mainly to the uncapped 5′ ppp-RNA byproduct[9,11,56]. We confirmed that RP-HPLC purification significantly reduced the amount of luciferase expression, which would reflect the removal of dsRNA impurities (Fig. 6a, ARCA, crude vs. ARCA)[33].

These experiments demonstrate the effect of removing 5′-triphosphate from uncapped mRNA, which is included as an impurity, on lowering the immunostimulatory effect of prepared mRNA. At the same time, they show the effectiveness of the PureCap method, which can remove both 5′-ppp-RNA and dsRNA impurities in the RP-HPLC purification process.

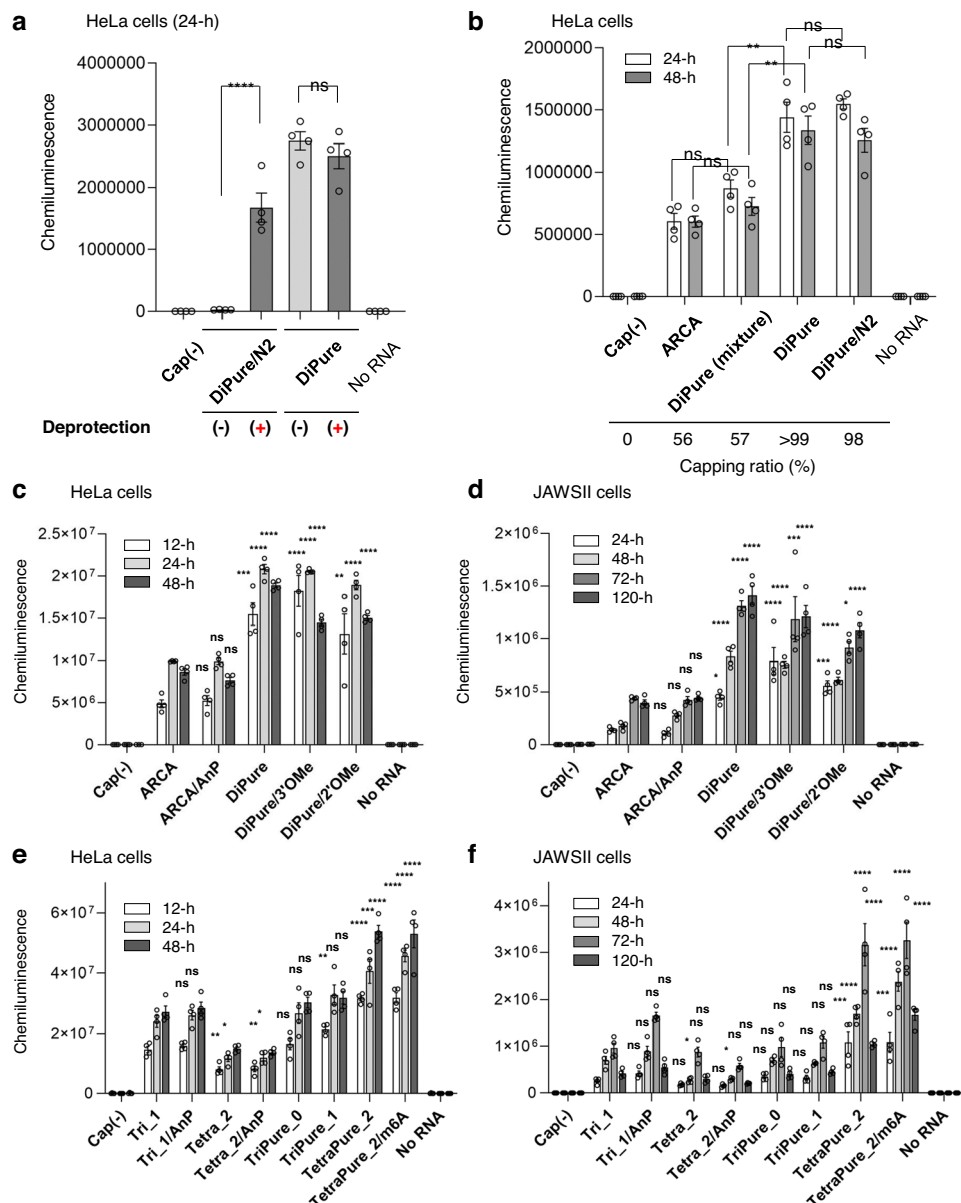

**Fig. 7 | Translation activities of Nluc mRNAs in cultured mammalian cells.** Capped mRNAs were introduced into HeLa cells (**a**–**c**, **e**) or JAWS II cells (**d**, **f**) using Lipofectamine MessengerMAX reagent. Cells were cultured for the indicated time, lysed, and the Nluc expression was measured. The cap analog used for mRNA preparation is indicated in the graph as the name of the mRNA. AnP means the mRNA was further treated with Antarctic phosphatase to remove its 5' phosphate. Data were mean ± s. e. m. for biological replicates ($n = 4$). **a**, **b** The statistically significant differences for the capped RNA samples in one-way ANOVA followed by Tukey's test were marked as follows. ns, $>0.05$; *$p < 0.05$; **$p < 0.01$; ***$p < 0.001$. **c**–**f** The statistically significant differences for the capped RNA samples from "ARCA" (**c**, **d**) or "Tri_1" (**e**, **f**) in one-way ANOVA followed by Dunnett's test were marked as follows. ns, $>0.05$; *$p < 0.05$; **$p < 0.01$; ***$p < 0.001$. **a**–**f** Each experiment was repeated independently twice (**a**, **b**) or at least three times (**c**–**f**) to obtain similar results. Source Data are provided with this paper.

## Translation activity measurement of Nluc mRNAs prepared by the PureCap method

First, Nluc mRNA was prepared using two dinucleotide PureCap analogs DiPure (**1**) and DiPure/N2 (**4**), in which Nb tags were introduced at different positions, and the effect of these tags on the translation activity of the mRNA was investigated (Fig. 7a). Capped Nluc mRNAs were isolated by RP-HPLC, and both samples were introduced into cultured HeLa cells derived from human cervical carcinoma before and after deprotection by light irradiation (Fig. 7a). It was found that the presence of Nb protecting group at $N^2$ of $m^7G$ base significantly suppressed mRNA translation, showing only 1.6% of the activity after deprotection. This observation is consistent with a recent report by Klocker et al. that the translation of mRNA containing a cap structure with a photocleavage group on the exocyclic amino group of $m^7G$

could be triggered by light irradiation in cultured cells[31,57]. They attributed the translation inhibition effect of the photocleavage group to the inhibition of binding to eIF4E[31]. On the other hand, unlike the case above with DiPure/N2, the mRNA prepared with DiPure showed almost the same translation activity before and after light irradiation, indicating that the Nb group at the 2' O-position does not impair the translation activity. We chose the hydroxyl group of the sugar moiety of the $m^7G$ as the position for introducing the Nb group as a purification tag because of the relative ease of synthesis and the possibility of introducing subsequent chemical modifications at the $N^2$ position of $m^7G$.

Next, we checked the effect of increasing capping efficiency on mRNA translation activity (Fig. 7b). We prepared a pair of mRNAs differing only in capping efficiency, with almost 100% in one and lower

**Table 1 | Summary of translation activity of Nluc mRNAs evaluated in this study**

| Name of mRNA[a] | Capping efficiency (%)[b] | Normalized Nluc expression[c] | | | |
|---|---|---|---|---|---|
| | | HeLa cells | JAWS II cells | Mice liver | Mice spleen |
| Dinucleotide Cap analogs ($m^7G$-ppp-G) | | | | | |
| ARCA | 56 | $1.00 \pm 0.02$ | $1.00 \pm 0.03$ | n.d. | n.d. |
| ARCA/ AnP | 56 | $0.97 \pm 0.03$ | $1.13 \pm 0.03$ | n.d. | n.d. |
| DiPure | >99 | $2.36 \pm 0.05$ | $3.51 \pm 0.09$ | n.d. | n.d. |
| DiPure/3'OMe | >99 | $2.27 \pm 0.07$ | $3.11 \pm 0.21$ | n.d. | n.d. |
| DiPure/2'OMe | >99 | $2.01 \pm 0.09$ | $2.58 \pm 0.08$ | n.d. | n.d. |
| Tri/ tetranucleotide cap analogs $m^7G$-ppp-AG(G) | | | | | |
| Tri_1 | 87 | $1.00 \pm 0.04$ | $1.00 \pm 0.05$ | $1.00 \pm 0.15$ | $1.00 \pm 0.13$ |
| Tri_1/ AnP | 87 | $1.07 \pm 0.03$ | $1.51 \pm 0.06$ | $1.47 \pm 0.23$ | $0.61 \pm 0.08$ |
| Tetra_2 | 52 | $0.52 \pm 0.02$ | $0.70 \pm 0.04$ | n.d. | n.d. |
| Tetra_2/AnP | 52 | $0.51 \pm 0.02$ | $0.53 \pm 0.02$ | n.d. | n.d. |
| TriPure_0 | >99 | $1.12 \pm 0.06$ | $1.04 \pm 0.08$ | n.d. | n.d. |
| TriPure_1 | >99 | $1.31 \pm 0.05$ | $1.07 \pm 0.05$ | $0.80 \pm 0.08$ | $0.90 \pm 0.07$ |
| TetraPure_2 | 98 | $1.91 \pm 0.06$ | $2.99 \pm 0.19$ | $4.86 \pm 0.61$ | $2.48 \pm 0.29$ |
| TetraPure_2/m6A | 95 | $1.98 \pm 0.07$ | $3.60 \pm 0.19$ | $4.07 \pm 0.48$ | $3.58 \pm 0.63$ |

[a]The.RNAs were named after the cap analog used. AnP means the RNA was further dephosphorylated using Antarctic phosphatase.

[b]These values are taken from Figs. 2f, 4g.

[c]These data are calculated from the data shown in Figs. 6c–f, 7, represented as means ± s. e. m. The activities were normalized to the control samples that were ARCA for mRNAs prepared with a dinucleotide cap analog or Tri_1 for mRNAs prepared with a tri/tetranucleotide cap analog. All data were considered for evaluation If the activity was measured at more than one time point.

in the other. Firstly, mRNA transcribed using DiPure cap analog (**1**) was divided into two groups. From one, capped mRNA was purified using RP-HPLC followed by photo-irradiation. This process provided mRNA with over 99% capping efficiency, which is denoted as DiPure. The other group of mRNA was photo-irradiated before RP-HPLC purification. This method provides a single peak in RP-HPLC containing the mixture of uncapped and capped mRNA, as photo-irradiation removes a hydrophobic tag from capped mRNA. The resulting mRNA mixture showed a capping efficiency of 57%, denoted as DiPure (mixture). DiPure and DiPure (mixture) originated from the same mRNA stock, possessed the same cap structure, and similarly received HPLC purification but differed only in capping efficiency. After the introduction to HeLa cells, DiPure (mixture) mRNA showed approximately 60% translation efficiency of that obtained by DiPure mRNA with 100% capping efficiency. The difference in capping efficiency may explain the difference in translation activity in this result.

Since the above experiments showed that the PureCap method of mRNA preparation is effective for obtaining the highest translation activity of mRNA, we evaluated the translation activity of Nluc mRNAs with various cap structures in more detail. Nluc mRNAs were transfected into HeLa cells, a representative of non-immune cells, and mouse immature dendritic cell-derived JAWS II cells, a representative of immune cells, to examine the effect of different cap structures on the translation activity of mRNA (Fig. 7c–f and Table 1). As shown in Fig. 7b, c, mRNA prepared with DiPure showed two to three times higher translation activity than mRNA prepared with ARCA in cultured HeLa cells. Dephosphorylation of the uncapped mRNA in ARCA-capped mRNA hardly changed the translation activity (ARCA vs. ARCA/AnP). Similar results were confirmed using JAWS II cells, where DiPure-capped mRNA showed approximately 3-fold more potent activity than ARCA-capped mRNA (Fig. 7d). These differences in translation activity between these two will be attributed to the difference in the capping efficiency. That is, DiPure-capped mRNA was quantitative, although ARCA-capped mRNA was about 60% (Fig. 3f). From these results, we confirmed that the preparation of quantitatively capped mRNA using PureCap analogs contributes to improving mRNA translation activity.

To investigate the effect of the methyl group introduced into the 2' or 3' hydroxyl of $m^7G$ on the translation activity, mRNAs prepared

with three PureCap analogs, DiPure, DiPure/2'OMe, and DiPure/3'OMe were compared (Fig. 7c, d, Table 1, and Supplementary Fig. 19a). In cultured HeLa and JAWS II cells, *O*-methylation at the 2' positions slightly reduced its translation activity. Still, no apparent change in the activity was observed when one methyl group was introduced into the cap structure[58]. These experimental results are consistent with in silico docking simulations showing that the methyl group introduced at the 3'-*O* position of $m^7G$ of dinucleotide cap analog would not interfere with the interaction of the cap-binding protein V39 of cap-specific mRNA 2'-*O*-methyltransferase or the eukaryotic translation initiation factor eIF4E[22]. In the end, we conclude that DiPure is the best of the three analogs because it is the easiest to synthesize, has the best incorporation by T7 RNAP, and exhibits high mRNA translation activity.

The translation activity of Nluc mRNAs with Cap-0, Cap-1, and Cap-2 cap structures was compared using cultured HeLa and JAWS II cells (Fig. 7e, f and Table 1). In these experiments, unlike previous reports[24], there was no apparent difference in activity between mRNAs with Cap-0 and Cap-1 in both HeLa and JAWS II cells. On the other hand, the translation activity of Cap-2-mRNAs introduced using the tetranucleotide PureCap analogs TetraPure_2 and TetraPure_2/m6A were higher than that of mRNAs with Cap-1 and Cap-0[27]. The enhanced translation activity of Cap-2 mRNA was more pronounced in JAWS II cells than in HeLa cells and was up to 4.2-fold more active than Cap-1 mRNA prepared with Tri-1 cap analog. The results of our experiment, in which the translation activity of mRNA containing the Cap-2 structure is higher than that having the Cap-1 structure in JAWS II cells, differ from the results of a study reported by ref. 27. Still, the reason for this difference is unclear at this point. For control mRNAs prepared from a DNA template containing the A-inserted promoter, the translation activity of Cap-1 mRNA prepared using the Tri_1 trinucleotide cap, which had a high capping efficiency of 90%[25], was comparable to that of mRNA prepared using the PureCap analog TriPure_1, both in HeLa and JAWS II cells. The activity of Cap-2 mRNA prepared with the tetranucleotide cap Tetra_2, in which only about 50% of the cap was introduced, was only about 1/3 of that of Cap-2 mRNA prepared with TetraPure_2. This result reflects the low capping efficiency of tetranucleotide cap analogs and indicates the PureCap method's usefulness

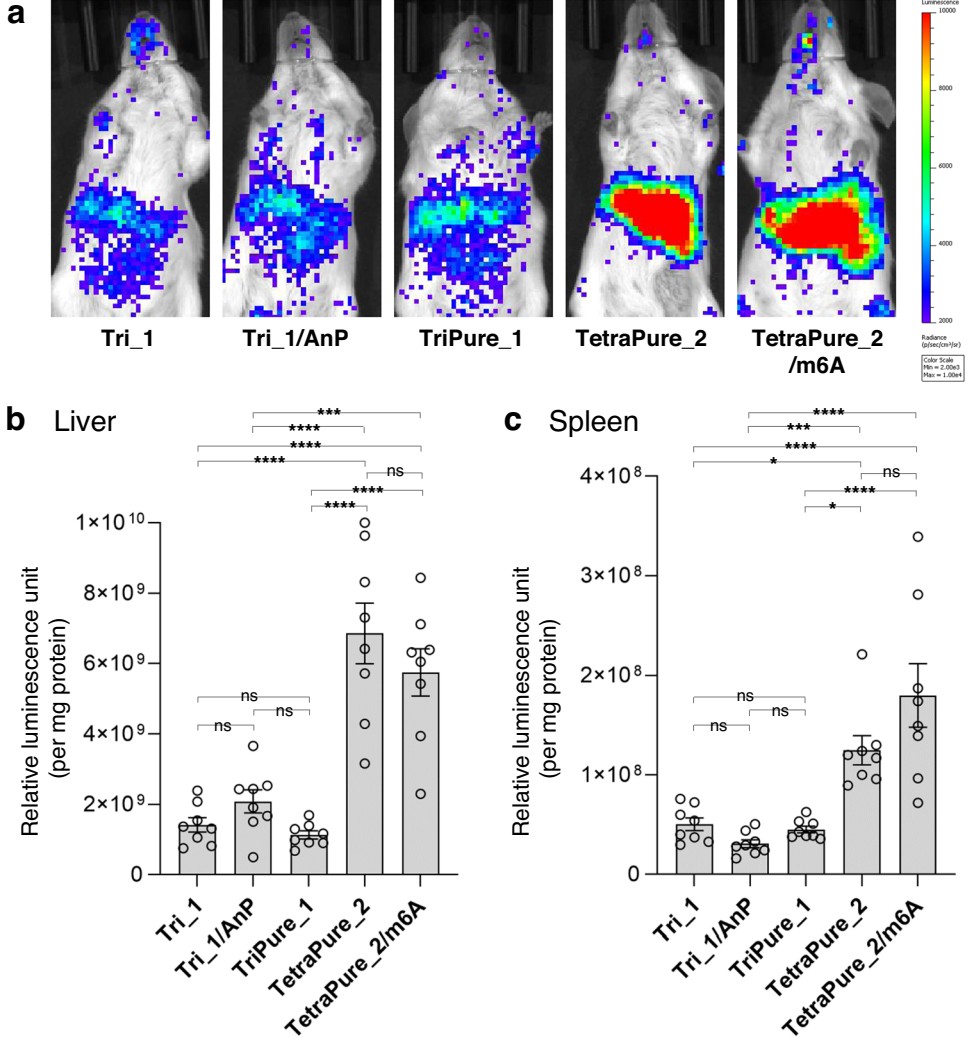

**Fig. 8 | Translation activity of 650-nt Nluc mRNA in mice.** The expression level of Nluc was evaluated four hours after intravenous injection of mRNA lipid nano-particles (LNPs) into mice. The cap analog used for mRNA preparation is indicated in the Figure. AnP means the mRNA was further treated with Antarctic phosphatase. (**a**) In vivo images of mice. (**b, c**) Quantifying Nluc expression levels using homogenates of the liver (**b**) and spleen (**c**). Data are mean ± s. e. m. for biological replicates ($n = 8$). The statistically significant differences in one-way ANOVA followed by Tukey's test were marked as follows: ns, $p > 0.05$ (not significant); *, $p < 0.05$; **, $p < 0.01$; ***, $p < 0.001$, ****, $p < 0.0001$. (**a–c**) Each experiment was repeated independently at least twice to obtain similar results. Source Data are provided with this paper.

in introducing Cap-2 structures using tetranucleotide cap analogs with low incorporation efficiency. As for Cap-2 mRNA containing m6A prepared using TetraPure_2/m6A, a slight increase when compared mRNA with TetraPure_2 in translation activity was observed in JAWS II cells.

We also compared the translation activity of mRNAs with this Cap-0, 1, and 2 structures in a cell-free translation system based on HeLa S3 cells[59]. In this experiment, Cap-2-mRNAs also showed higher translation activity than Cap-0/1 mRNA, especially the Cap-2 with m6A (Supplementary Fig. 19b).

In our experiments, since the Cap-2 cap structure was found to increase the translation activity of its mRNA compared to the Cap-1 cap structure in cultured cell lines, these translation activities were subsequently compared in animal experiments using mice (Fig. 8). mRNA was encapsulated into ionizable lipid-based lipid nanoparticles (LNPs) for systemic administration. In vivo luminescence imaging revealed that TetraPure_2 and TetraPure_2/m6A tended to show enhanced Nluc expression compared to the Cap-1-containing mRNA prepared with the PureCap analog Tri-Pure_1, or control compound Tri_1 (Fig. 8a). LNPs provided strong Nluc expression in the liver. At the same time, the expression in the spleen may also contribute to the luminescence from the left

mouse side. This observation led us to evaluate the Nluc expression level more quantitatively using the liver and spleen homogenates. As observed in the in vivo imaging, Cap-2-containing mRNA showed ~3-fold higher translation in the liver and spleen with statistical significance compared to the Cap-1-containing mRNA in Nluc quantification from the homogenates (Fig. 8b, c and Table 1). To the best of our knowledge, this result demonstrated the advantage of Cap-2-mRNA over Cap-1-mRNA in increasing protein expression efficiency in animals for the first time.

**Preparation/purification of longer mRNA by the PureCap method**

Using the PureCap method, we attempted to purify capped mRNA with a longer sequence. As a general trend, we found that the degree of separation decreased as the length of the mRNA increased. Therefore, we designed and synthesized more hydrophobic cap analogs (Fig. 9a, **23–25** and Supplementary Figs. 12–14). The substituent at the benzyl position of Nb was changed from the original *tert*-butyl to more hydrophobic phenylethyl, *n*-hexyl, or *n*-undecyl in the dinucleotide PureCap analog DiPure (**1**). We selected 4247-nt mRNA encoding the

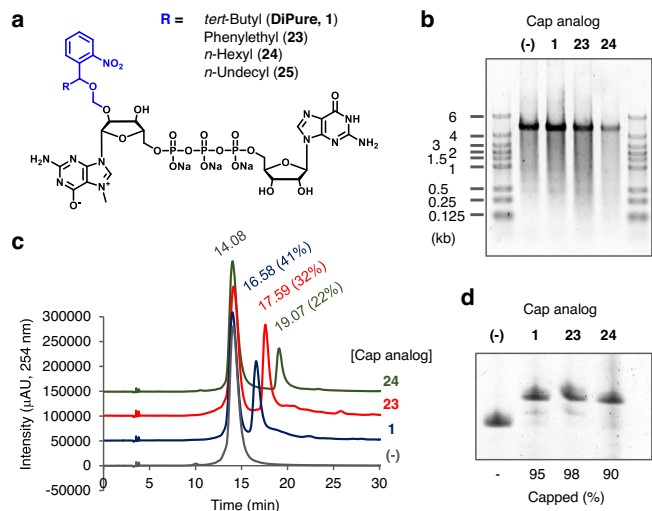

**Fig. 9 | Preparation and purification of capped 4247-nt mRNA encoding spike protein of SARS-CoV-2 using dinucleotide PureCap analogs. a** Chemical structure of the cap analogs with higher hydrophobicity (**23–25**) compared to the original dinucleotide analog (**1**). **b** Denaturing agarose gel electrophoresis of the IVT reactions in the presence of these cap analogs. **c** RP-HPLC analysis of IVT-RNAs with or without cap analogs. Elution time (min) and the ratio of capped mRNA were noted near the peak. **d** Analysis of 5′ terminus of mRNA after RP-HPLC purification. After being cleaved by DNAzyme 10–23, it was analyzed by 15% dPAGE. **b–d** Each experiment was repeated independently at least twice to obtain similar results. Source Data are provided with this paper.

spike protein of SARS-CoV2[3] as a target and tested if an increase in the hydrophobicity of the cap analog could improve the separation of capped mRNA. In a preliminary study, the most hydrophobic compound **25** formed a precipitate in the reaction mixture, producing almost no transcript (Supplementary Fig. 20). The other three analogs successfully provided RNA transcripts according to agarose gel electrophoresis, although compound **24** caused a slight decrease in transcription efficiency (Fig. 9b). This result suggests that highly hydrophobic cap analogs may hamper the transcription reaction of T7 RNAP. Meanwhile, cap analogs with increased hydrophobicity, compounds **23** and **24**, improved the separation of capped mRNA compared to the original compound **1** in RP-HPLC analysis, with a slight decrease in incorporation efficiency of cap analogs into mRNA (Fig. 9c). Notably, the proportion of a cap in the whole mRNA is 6.5-fold less in the spike mRNA compared to Nluc mRNA (one cap in 4247-nt in the spike mRNA and one cap in 650-nt in Nluc mRNA), making the separation of capped and uncapped mRNAs more difficult. However, the capped and uncapped long mRNAs were still separable with the original analog DiPure (**1**) and the cap analogs with enhanced hydrophobicity (compounds **23** and **24**). Capped mRNA was then separated twice by RP-HPLC, followed by 5′-end analysis using DNAzyme 10–23 (Fig. 9d). 95%, 98 and 90% of cap incorporation was achieved for compounds **1**, **23**, and **24**, respectively, after photo-cleaving and RP-HPLC purification.

## Discussion

Various cap analogs with different structural features have been developed to prepare capped mRNAs by IVT[19]. First to mention are the dinucleotide cap analogs ARCAs[21,22,39,58,60], which were designed to control the direction of introduction of the dinucleotide cap analog m$^7$G(5′)ppp(5′)G. It was reported that methylation of the 2′ or 3′ hydroxyl group completely controlled the direction of strand elongation, resulting in an approximately 2–3-fold increase in mRNA translation activity in cultured cells[58,60]. Following the successful development of ARCA, many modified analogs for dinucleotide caps

have been developed. Many examples of modification of the phosphate moieties have been developed[61,62]. Extensions of triphosphate linkage to tetra- and pentaphosphates were reported, and improved translation activity of capped mRNA prepared using these analogs was reported[39]. The development of analogs in which bridging or non-bridging oxygen atoms were replaced by S, Se, $CH_2$, and $BH_3$ was also reported[61–63]. Synthesis of cap analogs in which a non-phosphate linkage was introduced into the phosphate moiety by the click reaction has been reported[64,65]. These modifications in the linkage often increase the translation activity of the mRNA, mainly because they increase its resistance to degradation by the scavenger mRNA-decapping enzyme DcpS. Synthesis of functional cap analogs that can create labeled translatable mRNA by introducing fluorescent groups or biotin into the 2′ and 3′ hydroxyl groups of m$^7$G was also reported[63,66,67]. The introduction of a benzyl group at the $N^2$ position of m$^7$G or the substitution of a 7-methyl group with an arylmethyl group was reported to enhance the translation activity of mRNA in human cultured cells[58,68,69]. Some cap analogs have been developed for on/off control of the mRNA translation[31,70–72]. For example, Ogasawara et al. reported the synthesis of cap analogs that contained photo-responsible groups such as phyenylazo or styryl group or at the $N^2$ or $C^8$ position of m$^7$G and that their cis/trans conformation enabled photo-switchable translation control[70–72]. As the biological significance of the Cap-1 and Cap-2 structures becomes clearer, tri- and tetranucleotide cap analogs have also been developed to introduce them directly into mRNA[23–27].

Compared to the development of the cap structures mentioned above, much less effort has been devoted to improving capping efficiency[25,67]. Currently, a method to improve the incorporation efficiency of trinucleotide cap analogs exploiting their complementarity with the template is widely used[25]. Although the capping efficiency with this method is reported to be 80–90%[30], further improvement of the capping efficiency will be necessary because of the substantial amount of contaminating 5′ ppp-RNA that induces strong innate immunity[29]. As is well known, it is important to remove immunostimulatory dsRNA byproducts produced by T7 RNAP during transcription[33,34]. We undertook the present study to remove these byproducts for the therapeutic use of mRNA. The significance of this study lies in designing cap analogs for increasing capping efficiency, while previous studies derivatized numerous cap analogs for improving cap functionalities as mentioned above; the PureCap method we are developing here is unprecedented in that it removes 5′ ppp-RNA by chromatography. Indeed, we succeeded in completely suppressing innate immune responses without enzymatic treatment, such as Antarctic phosphatase (Fig. 6). Further notably, the hydrophobic tag introduced for purification can be removed simply by light irradiation, leaving no unnatural modification in the capped mRNA.

PureCap technology also provides a reliable platform to study structure-activity and the relationship of cap analogs. In co-transcriptional capping, different cap structures resulted in different capping efficiency (Fig. 5g), causing bias in their comparison. Indeed, our PureCap method achieved almost 100% capping efficiency in various capping structures, including Cap-0, Cap-1, and Cap-2, with or without modification in 2′ or 3′ hydroxyl of m$^7$G and $N^6$ position of the first transcribed A (m$^6$A). Especially, quantitative capping was obtained in preparing Cap-2 mRNA, which shows low capping efficiency in the conventional method (Fig. 5g). Cap-2 mRNA thus prepared showed up to three- to four-fold higher translation activity in cultured cells and animals than mRNA prepared by standard capping methods without inducing innate immune responses (Figs. 6–8). Intriguingly, Cap-2 mRNA improved protein translation efficiency by approximately threefold in the liver and spleen, two major target organs of mRNA therapeutics. Numerous studies and clinical developments target the liver for treating single gene disorders providing therapeutic proteins systemically[73,74], and the spleen for cancer vaccines[75],

immunotolerance vaccines[76], and chimeric antigen receptor T cell potentiation[77]. The Cap-2 mRNA technology is capable of potentiating these therapeutics.

One issue to be resolved in the future is the low incorporation efficiency of the tetranucleotide cap analogs to give the Cap-2 structure in mRNA, which was about 25% of the total. Currently, wild-type T7 RNA polymerase is used, but in the future, we would like to improve this, for example, with modification of the polymerase[78]. Scalability issues in the RP-HPLC purification process may be a concern. We will be able to learn from the process of the commercial-scale manufacturing of therapeutic oligonucleotides that have recently been launched[79]. Due to their structural complexity, these oligonucleotides are purified by HPLC even on a commercial scale[80–82]. We believe that mRNA purification above the gram scale using RP-HPLC is feasible using a system similar to that used for oligonucleotide therapeutics.

In summary, we synthesized quantitatively capped mRNAs by the PureCap method using synthesized hydrophobic cap analogs. To the best of our knowledge, our system provides a footprint-free purification method of capped mRNA with 100% capping efficiency for the first time. Notably, increasing the capping efficiency of mRNA resulted in increased translation activity and reduced immunostimulation. Practically, this technology requires only RP-HPLC for purifying capped mRNA, avoiding repeated enzymatic treatment commonly performed for reducing mRNA immunogenicity. Furthermore, this approach is versatile to apply to mRNA with different lengths (650 nt and 4247 nt). Achieving 100% capping efficiency in various cap species, our approach provides an unbiased platform to study their structure-activity relationship by excluding the influence of different capping efficiency observed among cap species. In this study, we prepared fully capped Cap-2 mRNA, which shows low capping efficiency in conventional methods. Ultimately, Cap-2 mRNA thus prepared showed up to three- to four-fold higher translation activity in cultured cells and animals than mRNA prepared by standard capping methods without inducing innate immune responses. Worth notably, Cap-2 mRNA improved protein translation efficiency in the liver and spleen, two major target organs of mRNA therapeutics, providing an excellent tool for future mRNA-based medicines.

## Methods

### Ethical statement
The animal experiments were conducted under the approval of the animal care and use committees at the Kyoto Prefectural University of Medicine and the Innovation Center of NanoMedicine (iCONM) with all relevant ethical regulations.

### Preparation of Nluc mRNA by IVT and its purification by RP-HPLC
The sequence of 650-nt Nluc mRNA is described in the Supplementary Information file. Template DNA for IVT was prepared by PCR reaction using pNL1.1 TK vector (Promega) as a template. The sequence of the PCR primers was as follows: Forward (Type III ϕ6.5 promoter-containing), 5′-CCCGGATCCTAATACGACTCACTATAGGCGCATATTAAGG TGACGCGT-3′; Reverse, 5′ (T)30-CTAGAATTACGCCAGAATGCG-3′. To prepare DNA containing different T7 promoters as Type II ϕ2.5 or the one named "A-inserted," a different forward primer was used as described below: For Type II ϕ2.5: 5′- CCCGGATCCTAATACG ACTCACTATTAGGCGCATATTAAGGTGACGCGT-3′: For "A-inserted"; 5′-CCCGGATCCTAATACGACTCACTATAAGGCGCATATTAAGGTGACG CGT-3′. The PCR mixture consisted of 0.5 µM primers, 1 ng/µL pNL1.1 TK vector, 0.2 mM dNTPs, 1.5 mM MgSO4, 1×PCR Buffer for KOD -Plus-Neo, 0.02 units/µL KOD -Plus- Neo (Toyobo). The mixture was subjected to the thermal cycling reaction as follows: 94 °C, 2 min → (98 °C, 10 s → 55 °C, 30 s → 68 °C, 90 s) × 25 cycles → 68 °C, 5 min. The reaction was analyzed by agarose gel electrophoresis, and dsDNA was recovered by alcohol precipitation after TE-saturated phenol/ chloroform

(1:1) extraction. A typical transcription reaction was carried out in a reaction mixture containing 15 ng/µL DNA (PCR product), 2 mM NTPs, 2 mM cap analog, 40 mM Tris-HCl (pH 8.0), 8 mM MgCl2, 2 mM spermidine, 5 mM DTT, 0.002 U/µL inorganic (yeast) pyrophosphatase (New England Biolabs), 9.4 ng/µL T7 RNA polymerase. For the control cap analogs, ARCA was purchased from Jena Bioscience, while Tri_1 and Tetra_2 were synthesized in-house. After the mixture was incubated at 37 °C for 2 h, DNase I (Takara bio) was added to the mixture at a final concentration of 0.1 units/µL, and it was further incubated at 37 °C for 15 min. After the mixture was extracted with TE-saturated phenol/ chloroform and chloroform, RNA was recovered by alcohol precipitation. Progress of the IVT reaction or mRNA purity after being purified was confirmed by denaturing PAGE analysis [5(w/v) % acrylamide (ratio of acrylamide and bis-acrylamide was 19:1), 7.5 M urea, 1× Tris-borate-EDTA (TBE) buffer (89 mM Tris, 89 mM boric acid, 2 mM EDTA, pH 8.3)]. The gel was stained with SYBR Green II and visualized on the ChemiDoc Touch MP imaging system (Bio-Rad). Nluc mRNA was analyzed and purified by RP-HPLC using YMC-TriartBio C4 column (250 × 4.6 mm I.D., S-5 µm, 12 nm) on Hitachi Chromaster (pump, 5110; detector 5430) or LaChrom (pump, L-2130; detector L-2455) HPLC system with Solution_A (50 mM triethylammonium acetate (TEAA, pH 7.0) containing 5% acetonitrile) and Solution_B (acetonitrile) at a flow rate of 1 mL/min. The content of Solution_B was raised from 0 to 20% over 20 min. The column temperature was maintained at 50 °C. After purification of mRNA by RP-HPLC, RNA was recovered by alcohol precipitation from the eluate in the presence of NaOAc (pH 5.2) and 2-propanol. RNA concentration was determined by measuring absorbance at 260 nm on NanoDrop 2000 spectrophotometer (Thermo). Nb-protection of the capped mRNAs prepared using PureCap analogs was removed by 365-nm light irradiation to the mRNA. mRNA solution was transferred into a well of the 96-well multi-well plate at a volume of 50 µL/well. 365-nm light was irradiated to the RNA at 4 mW/cm² for 10 min at room temperature using MAX-350 compact xenon light source (Asahi Spectra). After light irradiation, the progress of the deprotection reaction was confirmed by the change in the elution time in RP-HPLC analysis. The deprotected capped mRNA was purified by RP-HPLC under the same conditions as above to remove residual dsRNA contamination. RP-HPLC purified control mRNAs were dephosphorylated using Antarctic phosphatase (AnP, New England Biolabs) in a reaction mixture containing 250 ng/µL Nluc mRNA, 50 mM Bis-Tris-propane-HCl, 1 mM MgCl2, 0.1 mM ZnCl2 (pH 6), 0.25 U/µL AnP. After being incubated at 37 °C for 30 min, the mixture was extracted with TE-saturated phenol/ chloroform (1:1) and chloroform. RNA was then recovered by alcohol precipitation.

### Dot-blot analysis to detect dsRNA byproduct
mRNA samples were spotted onto the Amersham Hybond-N+ membrane and air-dried. After being blocked in TBT-T [50 mM Tris-HCl (pH 7.4), 150 mM NaCl, 0.05%(v/v) Tween-20] containing 5% (w/v) skim milk (Wako) for 1 h at room temperature, the blot was incubated with anti-dsRNA clone J2 (Sigma-Aldrich, MABE1134-100UL or Jena Bioscience, RNT-SCI-10010200) for 1 h at room temperature, which 1000-fold diluted with TBS-T containing 0.5% skim milk. After being washed with TBS-T, the blot was further incubated with anti-mouse IgG-HRP (Sigma-Aldrich, A9044) for 1 h at room temperature, diluted by 5000-fold with TBS-T containing 0.5% skim milk. After washing with TBS-T, the blot was incubated with SuperSignal West Femto Maximum Sensitivity Substrate (Thermo) and visualized on a ChemiDoc Touch MP imaging system (Bio-Rad). siRNA Ladder Marker (Takara, cat# 3430) was used as a positive control of dsRNA.

### RNA cleavage using a DNAzyme 10–23 to analyze the 5′ end of mRNA
DNAzyme 10–23 was designed according to a previous report[49]. The sequences used in this study are listed as follows (catalytic domain

underlined): for Nluc mRNA, 5′-TTCGAGGCCA<u>GGCTAGCTA-CAACGA</u>ACGCGTCACC-3′; for 34/35-nt short RNA, 5′-TTGTAGTC-<u>CAGGCTAGCTACAACGA</u>CGGATATATCTCCT-3′; for spike protein mRNA, 5′-TCTGTGGGGA<u>GGCTAGCTACAACGA</u>CAGAAGAATACTAG-3′. Target RNA (0.5 μM) and DNA (1 μM) were incubated in a buffer composed of 50 mM Tris-HCl (pH 8.0) and 50 mM $MgCl_2$ at 37 °C for 1 h. The mixture was then alcohol-precipitated. 1.57 μg RNA for 650-nt Nluc mRNA was digested in a 15 μL reaction and analyzed by 15(w/v)% denaturing PAGE containing 7.5 M urea and 1×TBE. The gel was stained with SYBR Gold and visualized on a ChemiDoc Touch MP imager. As for MALDI-TOF MS analysis of the reaction, the reaction mixture (10 μL) was twice alcohol-precipitated using ammonium acetate to remove sodium ions. It was measured in a positive mode with an Ultra-fleXtreme MALDI-TOF/TOF mass spectrometer (Bruker Daltonics) using 3-hydroxypicolinic acid as a matrix.

### Short RNA transcription system using 60-bp DNA as a template

IVT templates containing three different T7 RNA promoter sequences were prepared by the extension reaction of 26-nt DNA oligo annealed to the 60-nt oligo using DNA polymerase. The sequences of the oligos used in this experiment were listed below (5′→3′ direction; underlined promoter sequence; $N_m$, 2′-O-methyl-modified).Type III φ6.5 containing (60-nt), 5′-CCCGGATCC<u>TAATACGACTCACTATAG</u>GGATCCGAAG GAGATATATCCGATGGACTACAA-3′; Type II φ2.5 containing (60-nt), 5′-CCGGATCC<u>TAATACGACTCACTATTAG</u>GGATCCGAAGGAGATATAT CCGATGGACTACA-3′; A-inserted containing (60-nt), 5′-CCGGATCC-<u>TAATACGACTCACTATAA</u>GGGATCCGAAGGAGATATATCCGATGGAC TACAA-3′; Reverse strand for all (26-nt): 5′ $U_m$$U_m$GTAGTCCATCG GATATATCTCCTT-3′. 60-bp dsDNA IVT templates were prepared as follows. A reaction mixture composed of 2 μM DNA oligos (60-nt and 26-nt), 0.2 mM dNTPs, 1×PrimeSTAR HS Buffer, 0.025 U/μL PrimeSTAR HS DNA polymerase was incubated at 95 °C for 2 min, 55 °C for 1 min, and 72 °C for 60 min. It was extracted with TE-saturated phenol/chloroform (1:1) chloroform, and the DNA was alcohol-precipitated. Transcription of 34/35-nt RNA was carried out in a reaction mixture containing 15 ng/μL 60-bp dsDNA, 2 mM NTPs, 2 mM PureCap analog, 40 mM Tris-HCl (pH 8.0), 8 mM $MgCl_2$, 2 mM spermidine, 5 mM DTT, 9.4 ng/μL T7 RNA polymerase. After incubation at 37 °C for 2 h, a portion (0.5 μL) was taken from the mixture and analyzed by 15% denaturing PAGE containing 7.5 M urea. The gel was visualized by SYBR Green II staining.

### Immunogenicity measurement of mRNAs

NF-κB reporter (Luc)-HEK293 cells (BPS Bioscience, 60650) were grown in Dulbecco's modified Eagle's medium (DMEM; Wako) supplemented with 10% fetal bovine serum (FBS; Invitrogen) and 100 μg/ml of Hygromycin B at 37 °C under 5% $CO_2$ atmosphere. One day before transfection, the cells were seeded in a 48-well cell culture plate at $1.0 \times 10^5$ cells/well. Just before transfection, the medium was replaced with Opti-MEM I Reduced Serum Medium (240 μL/well, Thermo). mRNA (100 ng/well) was mixed with Lipofectamine MessengerMAX (0.15 μL/well) in Opti-MEM I medium (10 μL/well) and added to cells. Three hours after the transfection, DMEM containing 20% FBS was added to the cells (250 μL per well). Twenty-four hours after transfection, the cells were lysed using Glo Lysis Buffer (50 μL/well, Promega), and the luciferase expression was measured using ONE-Glo Luciferase Assay System (Promega). The chemiluminescence was measured on the TriStar5 plate reader (Berthold). The luciferase expression was normalized by total protein concentration in the lysate measured using Pierce BCA Protein Assay Kit (Thermo).

### Translation activity measurement of Nluc mRNA using cultured mammalian cells

HeLa cells (Riken Cell Bank, RCB0007) were grown in DMEM (Wako) supplemented with 10% FBS (Invitrogen) at 37 °C under a 5% $CO_2$

atmosphere. JAWS II cells (ATCC, CRL-11904) were grown in MEM α, nucleoside (Gibco) supplemented with 10% FBS (Invitrogen) and 5 ng/mL murine GM-CSF (PeproTech) at 37 °C under a 5% $CO_2$ atmosphere. One day before transfection, the cells were seeded in a 96-well cell culture plate, typically at $5 \times 10^3$ cells/well for HeLa cells or $1 \times 10^4$ cells/well for JAWS II cells. Just before transfection, the medium was replaced with Opti-MEM I medium (90 μL/well, Thermo). mRNA (10 ng/well) mixed with Lipofectamine Messenger-MAX (0.15 μL/well) in Opti-MEM I medium (10 μL/well) was added to the cells. Three hours after the transfection, the growing medium was added to the cells (100 μL per well). At indicated time points, the cells were lysed, and the luciferase expression was measured using Nano-Glo Luciferase Assay System (Promega). The chemiluminescence was measured on the TriStar5 or TriStar2 plate reader (Berthold Technologies). Statistical analysis of the data was performed using GraphPad Prism 9 software.

### Translation activity measurement of Nluc mRNA using HeLa S3 cell-based cell-free translation system

HeLa S3 cell extract was prepared based on a previous report[59]. HeLa S3 cells (Riken Cell Bank, RCB1525) were cultured in 1 L of S-MEM (Gibco) supplemented with 10% FBS (Corning) and 2 mM ʟ-glutamine (Nacalai) using a spinner flask (Corning) at 37 °C under 5% $CO_2$ atmosphere. When the cell number reached the order of the ninth power of 10, cultured cells were precipitated by centrifugation and washed three times with the washing buffer [35 mM HEPES-KOH (pH 7.5), 140 mM NaCl, and 11 mM glucose] and once with the extraction buffer [20 mM HEPES-KOH (pH 7.5), 45 mM potassium acetate, 45 mM KCl, 1.8 mM magnesium acetate, and 1 mM DTT]. Then, the cell pellet was suspended with an equal volume of the extraction buffer and still left in the pressure vessel (Parr Instrument company), kept under 500 psi of nitrogen gas for 30 min on ice. Cells were disrupted by the sudden release of nitrogen gas pressure dissolved in the cell suspension. After disruption, 1/300 volume of the high potassium buffer [20 mM HEPES-KOH (pH 7.5), 945 mM potassium acetate, 945 mM KCl, 1.8 mM magnesium acetate, and 1 mM DTT] was added to the lysate and mixed well. It was centrifuged three times at 4 °C to recover the supernatant as cell extract, being divided into aliquots and stored at −80 °C. Translation mixture composed of 0.5 μL of 30 ng/μL Nluc mRNA (650-nt), 4.35 μL of HeLa S3 cell extract, 3.25 μL of Mixure-2 [12.7 mM DTT, 5.42 mM magnesium acetate, 85.4 mM potassium acetate, 282 mM HEPES-KOH (pH 7.5)], 0.9 μL of Mixture-3 [12.6 mM ATP, 1.22 mM GTP, 200 mM creatine phosphate, 0.6 mg/mL creatine kinase, 0.08–1.2 mM 20-amino acids mixture], was incubated at 32 °C. The expression of Nluc was measured after 0.5-, 1-, 2-, 4-, and 6 h using Nano-Glo Luciferase Assay System (Promega). Statistical analysis of the data was performed using GraphPad Prism 9 software.

### Translation activity measurement of Nluc mRNA using mice

For lipid nanoparticle (LNP) preparation, 595 μg/mL of mRNA was dissolved in citrate buffer (pH 3.0) and lipid mixture from D-Lin-MC3-DMA (MedChemExpress), 1,2-distearoyl-sn-glycero-3-phospholine (DSPC, Wako), cholesterol (Sigma), and 1,2-dimyristoyl-rac-glycero-3-methoxypolyethylene glycol-2000 (DMG-PEG2000, Avanti) was dissolved in ethanol at D-Lin-MC3-DMA/DSPC/Sigma/DMG-PEG2000 molar ratio of 50/10/38.5/1.5 and total ethanol concentration of 35 mM. The citrate buffer and ethanol solution were mixed at a volume ratio of 2 to obtain a nitrogen/phosphate ratio of 5 using microfluidics (NanoAssemblr® Spark, Precision Nanosystems). The solvent was exchanged for PBS by diluting the solution 40-fold and concentrating LNP using Amicon Ultra-15−30K centrifugal units (Merck Millipore). About 2 μg of mRNA encapsulated in LNP was injected into Balb/c mouse (female, 7-week-old, Charles River Laboratories Japan, Inc. Housing conditions were maintained as follows: 12-h/12-h dark/light cycle, 23 ± 2 °C

temperature and 40 to 60% humidity). For in vivo imaging, 5 µg/ mouse of furimazine (Chem Shuttle) was injected from the mouse tail vein. Immediately after the injection, the mouse was observed using IVIS Spectrum SP-BFM-T1 (Perkin Elmer). The liver and spleen were excised at 4 h after the LNP injection for quantification using tissue homogenates. The organs were homogenized with Passive lysis buffer (Promega), followed by luciferase assay using Lumat3 LB9508 luminometer (Berthold Technologies) and Nano-Glo Luciferase Assay System (Promega). Statistical analysis of the data was performed using Kaleida Graph 5.0 software.

### Preparation and purification of spike protein mRNA of SARS-CoV-2

Reaction conditions in the transcription of capped 4247-nt mRNA encoding spike protein of SARS-CoV-2 using cap analogs were the same as that of 650-nt Nluc mRNA. IVT template was prepared by a PCR reaction using poly-$dT_{80}$-containing reverse PCR primer to introduce a poly-$A_{80}$ tail to mRNA. mRNA sequence is described in the Supplementary Information file. Transcribed mRNA was analyzed and purified by RP-HPLC using YMC-TriartBio C18 column (YMC corp., 250 × 4.6 mm I.D., TA30S05-2546PTH) on Shimadzu Prominence HPLC system (pump, LC-20AD; detector, SPD-M40) with Solution_A (100 mM TEAA (pH 7.0) containing 5% acetonitrile) and Solution_B (100 mM TEAA (pH 7.0) containing 50% acetonitrile) at a flow rate of 1 mL/min. The content of Solution_B was raised from 15 to 22% over 30 min. The column temperature was maintained at 50 °C.

### Statistics and reproducibility

No statistical method was used to predetermine the sample size. No data were excluded from the analyse. The experiments were not randomized. The Investigators were not blinded to allocation during experiments and outcome assessment.

### Reporting summary

Further information on research design is available in the Nature Portfolio Reporting Summary linked to this article.

## Data availability

All data generated or analyzed during this study are included in this published article, in the supplementary information file, or in the source data files. Source data are provided with this paper.

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

## Acknowledgements

This work was supported by Japan Science and Technology Agency (JST) [CREST, JPMJCR18S1 to H.A.) and Japan Agency for Medical Research and Development (AMED) (LEAP, JP21gm0010008 to H.A.). We are grateful for the support of the Chemical Instrumentation Facility of the Research Center for Materials Science, Nagoya University, to conduct the analyses. We thank R. Ogisu and T. Nishikawa for their technical assistance in the IVT experiments.

## Author contributions

M.I., N.A., and H.A. conceptualized and designed the study. M.I., Z.L., S.A., K.O., A.B., Z.M., M.T., T.I., P.L., K.K., and H.M. performed the chemical synthesis. N.A., Y.N., D. K., H.H., F.H., and S.U. designed and performed biochemical experiments, including mRNA preparation and cell experiments. S.U. designed and performed animal experiments. M.I., N.A., F.H., Y.K., S.U., and H.A. wrote the manuscript.

## Competing interests

H.A. and S.U. are cofounders of Crafton Biotechnology, and the company focuses on the development of mRNA therapeutics. An international patent application covering part of this work has been filed by JST and Nagoya University (inventor M.I., N.A., Z.L., Y.N., F.H., Y.K., H.A.; publication number WO/2023/282245). The remaining authors declare no competing interests.
