## [Peer Review File · Nature Communications]

REVIEWER COMMENTS

Reviewer #1 (Remarks to the Author):

The authors presented a method to help purified capped from uncapped in vitro transcripts by using cap analogs containing a hydrophobic group that allow resolution on reverse phase HPLC. The hydrophobic group is cleavage by UV irradiation such that the natural cap structures can be generated. The authors also highlight that the HPLC-based purification method can remove dsRNA side-product from IVT.

The development of a photocleavable tag incorporated into the 5' cap of synthetic RNA that can facilitate purification of capped from uncapped RNA is a significant development in synthetic mRNA. The authors also showed that in contrast to a recent report that added a photocleavable group at the N2 position, modification at the 2'O position of the guanosine cap does not affect translation activity of the synthetic mRNA. However, the authors may consider reframing some of the messages they are trying to communicate - see below.

My expertise is on the enzymology and assay development. I cannot comment on the chemistry and the chemical synthesis parts of the manuscript.

The biochemical data support the authors' claims that the hydrophobic photocleavable tag facilitate the purification of capped from uncapped in vitro transcripts in a shorter transcript (650 nt). However, as the authors showed, the resolution of the capped and uncapped form of a ~4200 nt transcript decreased. The drop in resolution with transcript length is expected because of the method's dependence on the increased hydrophobicity of the tag. Instead grossing through this limitation, the authors may suggest that tags with higher hydrophobicity can be developed for long transcripts.

The authors highlight that the fact that HPLC used to purify the capped and uncapped form of in vitro transcripts facilitates dsRNA removal. Although dsRNA can indeed be separated from ssRNA by reverse phase HPLC, the fact that dsRNA can be removed during photocaged-capped transcript purification is incidental. In fact, HPLC itself is not entirely desirable in mRNA manufacturing due to its capacity and throughput. The authors may suggest methods other than HPLC that can also be used to purify the capped and tagged transcripts from the uncapped ones and downplay the incidental advantage of dsRNA removal.

The authors found that capped trinucleotides with the cap-2 structure (TetraPure_2 and Tetra_2) had very low incorporation efficiency. Can the authors suggest possible reasons? Would the authors perform control experiments where Cap-0 or Cap-1 capped trinucleotides are evaluated? Is there a reason why

cap-2 dinucleotides (m7GpppAmGm, for example) was not synthesized and evaluated? The authors tested the Cap-2 m6A trinucleotide (TetraPure_2/m6A). Would the authors examine Cap-1 m6A dinucleotide (m7Gpppm6AmG)?

Other suggestions/comments:

Figure 1: Delete figure 1. Integrate Fig. 1A to Figure 2. Figure 1B is more suited for graphical abstract purpose than a standalone figure.

Figure 3(i) and figure 5(i): are the immunoblots mis-labeled? I don't understand them.

Figure 4: it is a great ground work to establish that the use of phi6.5 promoter, which normally initiates with a guanosine in phage T7, works as good as phi2.5 promoter that normally initiates with an adenosine for capped dinucleotides or trinucleotides initiating with an adenosine.

Extended Figure 5: It is a thorough effort to examine the 2'OMe and 3'OMe version of DiPure. However, DiPure 3'OMe is really ARCA plus the photocleavable hydrophobic tag at the 2'O position. Why is DiPure/3'OMe has a higher translation output than ARCA-capped transcript?

Extended figure 6: I think the data is important (as I highlighted above) and should appear with the main text.

Reviewer #2 (Remarks to the Author):

The paper describes the synthesis of cap-1 and cap-2 mRNA cap analogs and its biological applications. The authors develop a method to separate between capped and uncapped mRNAs by reverse phase HPLC that is based on hydrophobic photoinduced tag. However, the authors have failed to cite the previously established method in the area of capped mRNA isolation (Bioorg. Med. Chem. Lett. 2011, 21, 6131-6134 and Org. Biomol. Chem. 2012, 10, 8570-8574). The authors have claimed that higher translational activity has been achieved by removing highly immunogenic dsRNA, byproduct through HPLC. Although these results are interesting, it's no means novel as earlier papers revealed the same observations (Reference 23 and 25 in the manuscript). While this photoinduced tag technique provides capped mRNA with 100% capping efficiency, the comparison data between the capped mRNA and mixture of mRNA of various cap analogs (capped and uncapped mRNA) is missing. The paper would add more value if the translational activity of capped mRNA outperforms translational activity using mixture of mRNA. Given the isolation of capped mRNA through HPLC, it appears that the present technique limits in terms of scalability. The present version seems too thin to be published in Nature Communication. As such, the current manuscript lacks scientific merit and novelty and is not suitable for publication in Nature Communication.

REVIEWER COMMENTS

Reviewer #1 (Remarks to the Author):

[Comment_1]

The authors presented a method to help purified capped from uncapped in vitro transcripts by using cap analogs containing a hydrophobic group that allow resolution on reverse phase HPLC. The hydrophobic group is cleavage by UV irradiation such that the natural cap structures can be generated. The authors also highlight that the HPLC-based purification method can remove dsRNA side-product from IVT.

The development of a photocleavable tag incorporated into the 5' cap of synthetic RNA that can facilitate purification of capped from uncapped RNA is a significant development in synthetic mRNA. The authors also showed that in contrast to a recent report that added a photocleavable group at the N2 position, modification at the 2'O position of the guanosine cap does not affect translation activity of the synthetic mRNA. However, the authors may consider reframing some of the messages they are trying to communicate - see below.

My expertise is on the enzymology and assay development. I cannot comment on the chemistry and the chemical synthesis parts of the manuscript.

[Response_1]

Thank you very much for your time and efforts in reviewing our manuscript. We are also very grateful for your positive appraisal of our 5' cap analog system, facilitating the purification of capped mRNA using a photocleavable tag. Your comments are very constructive and allowed us to improve the quality of our manuscript by adding experimental data and revising the text as described in Responses_1-12. We sincerely hope this revision is satisfactory.

[Comment_2]

The biochemical data support the authors' claims that the hydrophobic photocleavable tag facilitate the purification of capped from uncapped in vitro transcripts in a shorter transcript (650 nt). However, as the authors showed, the resolution of the capped and uncapped form of a ~4200 nt transcript decreased. The drop in resolution with transcript length is expected because of the method's dependence on the increased hydrophobicity of the tag. Instead grossing through this limitation, the authors may suggest that tags with higher hydrophobicity can be developed for long transcripts.

[Response_2]

Thank you for pointing out a critical issue. Indeed, the longer the mRNA length, the more difficult it becomes to separate the peaks by a single hydrophobic tag attached to the cap structure. To solve this issue, we developed new cap analogs with higher hydrophobicity to better separate long mRNAs. Specifically, we synthesized three dinucleotide cap analogs by changing the substituent at

the benzyl position from *t*-butyl to a more hydrophobic substituent such as phenylethyl, *n*-hexyl or *n*-undecyl. The newly synthesized cap analogs showed higher hydrophobicity than the original cap analog with *t*-butyl group. Among them, two cap analogs showed improved separation of capped mRNA for 4.2 kb of spike protein mRNA, increasing the difference in the elution time between two peaks of uncapped and capped mRNA from 2.58 min to 3.47 and 5.05 min in RP-HPLC. To include the structure of the new cap analogs and the RP-HPLC result, we revised the original figure (Fig. 9) and presented it as Fig. 10 in this revised manuscript. In addition, the experimental details were described in the result section of “Preparation/purification of longer mRNA by the PureCap method”.

[Comment_3]

The authors highlight that the fact that HPLC used to purify the capped and uncapped form of *in vitro* transcripts facilitates dsRNA removal. Although dsRNA can indeed be separated from ssRNA by reverse phase HPLC, the fact that dsRNA can be removed during photocaged-capped transcript purification is incidental. In fact, HPLC itself is not entirely desirable in mRNA manufacturing due to its capacity and throughput. The authors may suggest methods other than HPLC that can also be used to purify the capped and tagged transcripts from the uncapped ones and downplay the incidental advantage of dsRNA removal.

[Response_3]

Thank you for your valuable comment from a manufacturing standpoint. Although we appreciate your concern about HPLC purification in mRNA manufacturing, please allow us to humbly state that HPLC is potentially scalable and is now applied in the commercial-scale manufacturing of oligonucleotide therapeutics. Indeed, Weldon *et al.* have shown that the purification of a GalNAc-conjugated oligonucleotide can be increased to 2,270 g per day by improving the purification process using continuous HPLC with a twin-column (Weldon R. *et al.*, *J Chromatogr A*, 2022). On the other hand, we sincerely recognize that the PureCap method using HPLC requires further development for large-scale mRNA production. Accordingly, we added the following statement to the discussion section of the revised manuscript regarding the scalability of RP-HPLC-based mRNA purification, with some citations: “Scalability issues in the RP-HPLC purification process may be a concern. We will be able to learn from the process of the commercial-scale manufacturing of therapeutic oligonucleotides that have recently been launched. Due to their structural complexity, these oligonucleotides are purified by HPLC even on a commercial scale. We believe that mRNA purification above the gram scale using RP-HPLC is feasible using a system similar to that used for oligonucleotide therapeutics”.

As you pointed out, we recognize that the isolation of capped mRNA and removal of dsRNA byproduct are separate issues. In the revised manuscript, we have corrected carefully the sentences

that seem to conflate these two. We recognize that dsRNA impurity can be removed by RP-HPLC, but an alternative method such as utilizing cellulose powder has also been developed, as we described in the manuscript. Ultimately, it would be best if dsRNA generation in the IVT could be completely avoided. As an example aimed in that direction, it has recently been reported that the generation of dsRNA can be avoided by modifying RNA polymerase (Dousis A. et al, *Nat. Biotechnol.*, 2022), which now cited as ref. 37 in the revised manuscript. The essence of the PureCap method reported here is to separate capped and uncapped mRNAs to maximize their translation activity and suppress the immune response derived from the 5' triphosphate. For that purpose, cap analogs with a hydrophobic tag have been developed, and their physical properties make the use of reverse-phase HPLC the most effective. If the capped mRNA is prepared using another chemical modification such as biotin, affinity chromatography using streptavidin-biotin interaction would allow separation, as reported in reference 67. In our opinion, the cost and scalability issues of such affinity chromatography will be greater than those of RP-HPLC. In addition, prepared mRNA having a biotin tag on the cap structure is unsuitable for the next biological or medical application because the modification decreased translation ability. Therefore, the footprint-free preparation method, like our PureCap, is useful.

[Comment_4]

The authors found that capped trinucleotides with the cap-2 structure (TetraPure_2 and Tetra_2) had very low incorporation efficiency. Can the authors suggest possible reasons?

[Response_4]

Thank you for pointing out an important issue. Consistent with our experimental results, a patent application from TriLink corp. shows that the tetranucleotide cap analog m7G-ppp-Am-Gm-G had lower incorporation efficiency when compared to the trinucleotide cap analog m7G-ppp-Am-G (Pub. No. US 2018/0273576 A1 (Sep. 27, 2018), Figs 12H, 13A, 50% vs 99%). The reason for this low incorporation efficiency of the tetranucleotides is unknown but is possibly due to the instability of T7 RNA polymerase initiation complex loading a bulky tetranucleotide cap analog. In the crystal structure of the T7 RNA polymerase initiation complex loading triguanosine (ppp-G-G-G), only the first and second guanosine from the 3' end form Watson-Click base pair with a template strand (Cheatham, G. M. T. and Steitz, T. A., *Science* 1999). The third guanosine, having only a single hydrogen bond with the cytosine on the template, peels off from the transcript-template strand. This observation suggests that tetranucleotide cap analogs for Cap2 mRNA preparation, possessing tetranucleotide sequences complementary to the template may be less stable inside the initiation complex than di- and tri-nucleotide cap analogs for Cap0 and Cap1 mRNA preparation. Accordingly, we have added the following speculation in the result section of the revised manuscript to explain the low incorporation efficiency of the tetranucleotide cap analogs as follows; “The lower

incorporation efficiency of the tetranucleotide cap analogs regardless of the promoter type might be attributed to their steric bulkiness, hampering stable initiation complex formation between the tetranucleotide, template DNA, and polymerase.”

[Comment_5]

Would the authors perform control experiments where Cap-0 or Cap-1 capped trinucleotides are evaluated?

[Response_5]

We have not synthesized tetranucleotide cap analogs with the Cap-0 or Cap-1 structures because smaller trinucleotide cap analogs are sufficient for preparing Cap-0 and Cap-1 mRNA. Although the tetranucleotides are useful in introducing the Cap-2 structure directly into the mRNA, the incorporation efficiency of the tetranucleotide cap analogs into mRNA was relatively low, as answered in your Comment_3. Therefore, we solely used trinucleotide caps to prepare Cap-0 or Cap-1 mRNA.

[Comment_6]

Is there a reason why cap-2 dinucleotides (m7GpppAmGm, for example) was not synthesized and evaluated?

[Response_6]

Thank you for the valuable question. We have not synthesized m7GpppAmGm, which cannot be incorporated into mRNA during in vitro transcription. When the 2' hydroxyl group of the 3' terminal base of the cap analog is methylated, T7 RNA polymerase cannot elongate the RNA strand from the adjacent 3' hydroxyl group. This so-called anti-reverse activity provides a rationale for the development of the ARCA cap analog. This issue is stated in the revised manuscripts as follows; “Methylation of the terminus 3' hydroxyl of the trinucleotide cap analogs cannot introduce Cap-2 structure into mRNA, as is known as the rationale for the development of ARCA. To this end, tetranucleotide cap analogs are required to introduce Cap-2 structure in mRNA, as reported recently by Drazkowska *et al.*”

[Comment_7]

The authors tested the Cap-2 m6A trinucleotide (TetraPure_2/m6A). Would the authors examine Cap-1 m6A dinucleotide (m7Gpppm6AmG)?

[Response_7]

Thank you for your valuable suggestion. We have not synthesized the trinucleotide PureCap analog containing m6A for Cap-1 mRNA preparation, because a previous study reported the influence of methylation at position N^6 of adenosine in the Cap-1 structure (*NAR* 2020, 48, 1607).

Interestingly, the structure enhanced mRNA translation activity (*NAR* 2020, 48, 1607), which motivated us to introduce m⁶A into the Cap-2 structure. In contrast to the previous report of Cap-1 mRNA, the methylation of adenosine at N⁶ using TetraPure_2/m6A showed minimal influence on translation activity in both cultured cells and mice.

[Comment_8]

Other suggestions/comments:

Figure 1: Delete figure 1. Integrate Fig. 1A to Figure 2. Figure 1B is more suited for graphical abstract purpose than a standalone figure.

[Response_8]

We sincerely appreciate your suggestion. However, our understanding is that *Nature communications* do not provide an independent graphical abstract. Therefore, we presented the concept figure as figure 1B. In addition, the manuscript needs Figure 1A to provide background information to general readers to explain the cap structures. As both Figure 1A and B play an introductory role in this manuscript, we combined them to prepare Figure 1.

[Comment_9]

Figure 3(i) and figure 5(i): are the immunoblots mis-labeled? I don't understand them.

[Response_9]

We apologize for the unclear figures. As you pointed out, Fig 3i and Fig 5i show the results of immunoblotting with anti-dsRNA J2 antibody, revealing the successful removal of dsRNA from mRNA samples after RP-HPLC purification. In detail, two consecutive RP-HPLC purifications cleared dsRNA from all samples almost completely. Accordingly, the figures have been revised for clarity. The numbers beside each spot in the figure correspond to the number listed in the figure legend.

[Comment_10]

Figure 4: it is a great groundwork to establish that the use of phi6.5 promoter, which normally initiates with a guanosine in phage T7, works as good as phi2.5 promoter that normally initiates with an adenosine for capped dinucleotides or trinucleotides initiating with an adenosine.

[Response_10]

We are honored by this comment and are aware of the excellent work on trinucleotide cap analogs reported by Henderson *et al.* (Ref. 25: inserting bases with complementarity with cap analog into the phi6.5 promoter), but the PureCap method isolates capped mRNA from a mixture of the transcript. Therefore, in this study, we considered it important to maximize the yield of capped mRNA produced, not the capping efficiency. This is why we chose the phi6.5 promoter in this study.

[Comment_11]

Extended Figure 5: It is a thorough effort to examine the 2'OMe and 3'OMe version of DiPure. However, DiPure 3'OMe is really ARCA plus the photocleavable hydrophobic tag at the 2'O position. Why is DiPure/3'OMe has a higher translation output than ARCA-capped transcript?

[Response_11]

Thank you for the critical question. As you pointed out, DiPure/3'OMe is ARCA with a photodegradable tag. However, capping efficiency was different between mRNA prepared from DiPure/3'OMe and ARCA, with 100% in the former and approximately 60% in the latter. The difference can influence translation activity. To clarify this, their capping efficiency was shown in the figure legend of Extended Fig. 5, as follows: "The capping efficiency of these mRNAs, reproduced from Figs 3f and 5g, were as follows: **ARCA**, 56%; **DiPure**, >99%; **DiPure/3'OMe**, >99%; **DiPure/2'OMe**, >99%; **Tri_1**, 87%, **TriPure_0**, >99%; **TriPure_1**, >99 %; **TetraPure_2**, 98 %, **TetraPure_2/m6A**, 95 %."

[Comment_12]

Extended figure 6: I think the data is important (as I highlighted above) and should appear with the main text.

[Response_12]

We agree with you about the importance of this result. Indeed, the experimental results guided the subsequent molecular design. According to the comment, we have moved the data to the main text in the revised manuscript and presented it as Figure 7a.

We sincerely appreciate all your advice to improve our manuscript and hope that the revised manuscript will satisfy all of your concerns.

Yours sincerely,
Hiroshi Abe

REVIEWER COMMENTS

Reviewer #2 (Remarks to the Author):

[Comment_1]

The paper describes the synthesis of cap-1 and cap-2 mRNA cap analogs and its biological applications. The authors develop a method to separate between capped and uncapped mRNAs by reverse phase HPLC that is based on hydrophobic photoinduced tag. However, the authors have failed to cite the previously established method in the area of capped mRNA isolation (Bioorg. Med. Chem. Lett. 2011, 21, 6131-6134 and Org. Biomol. Chem. 2012, 10, 8570-8574).

[Response_1]

Firstly, thank you very much for your time and efforts in reviewing our manuscript. Your comments are very constructive and allowed us to improve the quality of our manuscript. We also appreciate your kind introduction of valuable references. The former article was cited in the original manuscript as reference 62, and appears as ref. 67 in the revised manuscript. We newly cite the latter article entitled "Dinucleotide cap analogue affinity resins for purification of proteins that specifically recognize the 5' end of mRNA", describing the purification of cap-binding proteins using a cap analog, as ref. 63 in the revised manuscript.

[Comment_2]

The authors have claimed that higher translational activity has been achieved by removing highly immunogenic dsRNA, byproduct through HPLC. Although these results are interesting, it's no means novel as earlier papers revealed the same observations (Reference 23 and 25 in the manuscript).

[Response_2]

As you kindly pointed out, the benefit of RP-HPLC purification to remove dsRNA has been widely established, as stated in the introduction section of the manuscript with citations, as follows. "Especially, purification of mRNA by RP-HPLC is an established method for eliminating double-stranded RNA (dsRNA) contaminants produced during the IVT, which increases the immunogenicity of mRNA and inhibits its translation". The essential benefit of the PureCap method reported here is the removal of immunostimulatory and non-functional uncapped 5' triphosphorylated mRNA from capped mRNA. Therefore, the previous papers you mention do not compromise the novelty of our study. Please check answers to your comments 3 and 4 for the benefit of increasing capping efficiency, and an answer to your comment 6 for the novelty of this study.

[Comment_3]

While this photoinduced tag technique provides capped mRNA with 100% capping efficiency, the comparison data between the capped mRNA and mixture of mRNA of various cap analogs (capped and uncapped mRNA) is missing.

[Response_3]

Thank you for the critical comments related to the benefit of increasing the capping efficiency. To evaluate the influence of capping efficiency on translation activity, we prepared three sets of Nluc mRNA possessing identical cap structure with HPLC purification but differing in capping efficiency: (1) mRNA prepared from ARCA and DiPure/3'OMe, (2) mRNA from Tri_1 and TriPure_1, and (3) mRNA prepared from Tetra_2 and TetraPure_2. The capping efficiency was 57% in ARCA-capped mRNA from ARCA, 87 % in Tri_1-capped mRNA, and 52% in Tetra_2-capped mRNA (Figs 3f and 5g), while PureCap method provided almost 100 % capping efficiency in all listed cap analogs. After the introduction to HeLa cells, PureCap mRNA showed enhanced protein translation activity than its counterpart mRNA possessing identical capping structure at lower efficiency, prepared from conventional capping methods (Fig 8, Extended Data Fig 5 in the revised manuscript). Notably, the enhancement of the translation activity was prominent for DiPure/3'OMe and TetraPure_2, presumably because their counterpart capping analogs, ARCA and Tetra_2, provided relatively low capping efficiency (57% and 52%, respectively).

[Comment_4]

The paper would add more value if the translational activity of capped mRNA outperforms translational activity using mixture of mRNA.

[Response_4]

Thank you again for the constructive comments related to the core concept of our manuscript. An answer to your comment 3 showed the comparison of mRNA translation activity between mRNA with 100% and lower capping efficiency. For more stringent comparison, we prepared a pair of mRNAs differing only in capping efficiency, with almost 100% in one and lower in the other. For this purpose, mRNA in vitro transcribed using DiPure was divided into two groups. From one, capped mRNA was purified using RP-HPLC followed by photo-irradiation. This process provided mRNA with over 99% capping efficiency, which is denoted as DiPure. The other group of mRNA was photo-irradiated before RP-HPLC purification. This method provides a single peak in RP-HPLC containing the mixture of uncapped and capped mRNA, as pre-photo-irradiation removes a hydrophobic tag from capped mRNA. The resulting mRNA mixture showed a capping efficiency of 57%, denoted as DiPure (mixture). DiPure and DiPure (mixture) originated from the same mRNA stock, possessed the same cap structure, and similarly received HPLC purification but differed only in capping efficiency. After introduction to HeLa cells, DiPure (mixture) mRNA showed approximately 60% translation efficiency of that obtained by DiPure mRNA with 100% capping

efficiency. The difference in capping efficiency may explain the difference in translation activity in this result. In the revised manuscript, we have added the new experimental result as Figure 7b.

[Comment_5]

Given the isolation of capped mRNA through HPLC, it appears that the present technique limits in terms of scalability.

[Response_5]

We are well aware of the technical difficulties of using RP-HPLC to purify mRNA above the gram scale. However, there are many researchers who wish to prepare mRNA free of impurities as much as possible and test the potential of mRNA medicine. For these research purposes, mg-scale preparations are often sufficient. In such cases, mRNA preparation by RP-HPLC is effective enough with our current technology. Now, several oligonucleotide drugs have recently been launched. Due to the complexity of their structure, oligonucleotide drugs are purified using HPLC in commercial-scale manufacturing. Indeed, Weldon *et al.* have reported that the purification of a GalNAc-conjugated oligonucleotide can be increased to 2,270 g per day by improving the purification process using continuous HPLC with a twin-column (*J Chromatogr A* 1663, 462734, 2022). By learning from their process, we believe that it is possible to purify mRNA on a gram scale using RP-HPLC. The following statements regarding the scalability issues in RP-HPLC purification process have been added to the revised manuscript, along with citations. “Scalability issues in the RP-HPLC purification process may be a concern. We will be able to learn from the process of the commercial-scale manufacturing of therapeutic oligonucleotides that have recently been launched. Due to their structural complexity, these oligonucleotides are purified by HPLC even on a commercial scale. We believe that mRNA purification above the gram scale using RP-HPLC is feasible using a system similar to that used for oligonucleotide therapeutics.”

[Comment_6]

The present version seems too thin to be published in Nature Communication. As such, the current manuscript lacks scientific merit and novelty and is not suitable for publication in *Nature Communication*.

[Comment_6]

Please allow us to highlight the important achievements of our study, listed as bullet points below.

1. Preparation of fully capped mRNA. Current mRNA capping techniques, including co-transcriptional and enzymatic capping, provide a maximum of around 90% capping efficiency. Notably, uncapped and capped mRNA possesses almost identical physicochemical properties, posing challenges to the physical separation of these two products. Herein, we developed hydrophobic photocaged tag-modified cap analogs, which allowed purification of capped mRNA

simply by reversed-phase HPLC and recovery of native cap structure by photo-irradiation. To the best of our knowledge, our system provides a footprint-free purification method of capped mRNA with 100% capping efficiency for the first time.

2. Demonstrating the benefits of increasing capping efficiency. As described to the answers to your comments 3 and 4, fully capped mRNA outperformed partially capped mRNA prepared by conventional methods in translation activity in cultured cells. Moreover, increasing capping efficiency contributed to suppressing immunostimulation after introduction into cultured cells (Figure 6). These data demonstrate the benefit of our system to deliver mRNA efficiently without immunostimulation. Further, this approach is versatile to apply to a reporter mRNA (650 nt) to SARS-CoV2 spike mRNA (4,247 nt). Please also check the newly added Figure 10 for the latter.

3. Cap-2 mRNA preparation. Our approach enables preparing fully capped Cap-2 mRNA, while cap-2 mRNA preparation is challenging in the conventional method, providing approximately 50% capping efficiency. Cap-2 mRNA thus prepared showed up to 3 to 4-fold higher translation activity in cultured cells and animals than Cap-1 mRNA. Especially, this study demonstrated the utility of Cap-2 mRNA in an animal for the first time. Further notably, a recent report reveals endogenous functionalities of Cap-2 structure, drastically reducing the mRNA affinity to RIG-I, an innate immune receptor recognition compared to Cap-1 structure, with moderately increasing mRNA stability and translation activity (*Nature* 2023). This report provides an additional rationale to Cap-2 mRNA preparation in mRNA therapeutics.

To show a rationale for preparing Cap-2 mRNA, we added the description of endogenous identity and functionalities of Cap-2 mRNA in the introduction section of the revised manuscript as follows; "A recent study provided insight into the identity and functions of Cap-2 cap structure in mRNA, which remained unknown for long years. Cap-2 structure drastically reduced the mRNA affinity to retinoic acid-inducible gene-1 (RIG-1), an innate immune receptor recognition compared to Cap-1 structure, with moderately increasing mRNA stability and translation activity."

We sincerely appreciate all your advice to improve our manuscript and hope that the revised manuscript will satisfy all your concerns.

Yours sincerely,

Hiroshi Abe

REVIEWERS' COMMENTS

Reviewer #1 (Remarks to the Author):

I am glad that the authors found my comments helpful. I have the following comments on the revised manuscript submitted. I especially found the data shown in Fig. 10 on the benefit of more hydrophobic groups compelling for publishing in Nature Communications. I also appreciate that the authors mentioned negative results of compound 25 in the text and share the data in the form of supporting fig.

Comment_6/response_6 and line 189:

I think the authors mixed up 2'-O-Me and 3'-O-Me on ARCA. Cap-2 has a 2'-O-me on the second nucleotide; ARCA has a 3'-O-Me group. My question would be, would a m7GpppAmGm (both being 2'-O-Me) can be incorporated by T7RNAP?

Lines 102 and 103: 100% efficiency – change to “100% capping efficiency.”

Line 824-826: change “The resulting capping efficiency was 95%, 98%, 90% for compounds 1, 23, 24, respectively, showing successful purification of long capped mRNA.” to “95%, 98% and 90% of cap incorporation was achieved for compounds 1, 23, 24, respectively, after photo-cleaving and RP-HPLC purification.” The way it was written implies that the compounds generated 90+% of capped RNA in IVT, which is not true. For the sake of consistency, the authors should label the peaks in the chromatographs with percentage of capped species in Fig. 10c as in Fig. 3 and 5.

Reviewer #2 (Remarks to the Author):

Thanks to the authors for revising the manuscript. The authors have addressed our queries. In general, trinucleotide cap analogs without photocleavable tag provide around 95% capping efficiency (clean cap analog) but the present cap analogs with photocleavable tag provide 100% capping efficiency. In terms of translational efficiency, the translational properties of both trinucleotide with and without photocleavable tag are comparable, whereas in the case of tetranucleotide analog, photocleavable tag provides 3 to 4-fold higher translational efficiency. One of the major areas to improve the readability of the manuscript is to prepare a table summarizing the capping efficiency and translational data so that readers can easily understand the outcome of the results. The present findings provide marginal

improvements and does not provide any major breakthrough in the area of mRNA cap analogs in order to attract the Nature Communications audience. As such, the present revised manuscript is still too thin w.r.t. to the translation outcome but the method has some merit with revised clarity.

REVIEWERS' COMMENTS

Reviewer #1 (Remarks to the Author):

[Comment_1]

I am glad that the authors found my comments helpful. I have the following comments on the revised manuscript submitted. I especially found the data shown in Fig. 10 on the benefit of more hydrophobic groups compiling for publishing in Nature Communications. I also appreciate that the authors mentioned negative results of compound 25 in the text and share the data in the form of supporting fig.

[Response_1]

Thank you very much for your time and efforts in reviewing our revised manuscript. We are pleased to read your comment that our efforts to compile the data presented in Figure 10 and Supporting Figure S1 were especially appreciated. We have carefully considered your comments in this revision and have done our best to improve the manuscript. We sincerely hope this revision is satisfactory.

[Comment_2]

Comment_6/response_6 and line 189:

I think the authors mixed up 2'-O-Me and 3'-O-Me on ARCA. Cap-2 has a 2'-O-Me on the second nucleotide; ARCA has a 3'-O-Me group. My question would be, would a m7GpppAmGm (both being 2'-O-Me) can be incorporated by T7RNAP?

[Response_2]

We apologize for the lack of clarity in our explanation. We now prepared a figure to explain the effect of methylation of 2' or 3' hydroxyl of the 3' terminal nucleotide for the anti-reverse activity. The cap analog m⁷GpppAmGm (both being 2'-O-methylated, as shown in **d** in this figure) would not be incorporated by T7 RNAP in the right orientation, due to the presence of adjacent 2'-O-Me next to the 3' hydroxyl. The point is that methylation of the 2' hydroxyl also inhibits the elongation of the strand from the adjacent 3' hydroxyl. It was reported in 2003 by the excellent pioneering work by Jemielity *et al.*, which were cited as #39 in the manuscript.

[Comment_3]

Lines 102 and 103: 100% efficiency – change to “100% capping efficiency.”

[Response_3]

Thank you for your valuable suggestion. We have corrected the wording in line 102 and 103; “100% efficiency” was changed to “100% capping efficiency.”

[Comment_4]

Line 824-826: change “The resulting capping efficiency was 95%, 98%, 90% for compounds 1, 23, 24, respectively, showing successful purification of long capped mRNA.” to “95%, 98% and 90% of cap incorporation was achieved for compounds 1, 23, 24, respectively, after photo-cleaving and RP-HPLC purification.” The way it was written implies that the compounds generated 90+% of capped RNA in IVT, which is not true. For the sake of consistency, the authors should label the peaks in the chromatographs with percentage of capped species in Fig. 10c as in Fig. 3 and 5.

[Response_4]

Thank you for your valuable suggestions. The sentence in lines 824-826 was changed from “The resulting capping efficiency was 95%, 98%, 90% for compounds **1, 23, 24**, respectively, showing successful purification of long capped mRNA” to “95%, 98% and 90% of cap incorporation was achieved for compounds **1, 23, 24**, respectively, after photo-cleaving and RP-HPLC purification”. For the display of the percentage of capped RNA species, numbers are shown above the peaks as in Figures 3 and 5.

We sincerely hope this revision is satisfactory to you.

Hiroshi Abe

REVIEWERS' COMMENTS

Reviewer #2 (Remarks to the Author):

[Comment_1]

Thanks to the authors for revising the manuscript. The authors have addressed our queries. In general, trinucleotide cap analogs without photocleavable tag provide around 95% capping efficiency (clean cap analog) but the present cap analogs with photocleavable tag provide 100% capping efficiency. In terms of translational efficiency, the translational properties of both trinucleotide with and without photocleavable tag are comparable, whereas in the case of tetranucleotide analog, photocleavable tag provides 3 to 4-fold higher translational efficiency. One of the major areas to improve the readability of the manuscript is to prepare a table summarizing the capping efficiency and translational data so that readers can easily understand the outcome of the results. The present findings provide marginal improvements and does not provide any major breakthrough in the area of mRNA cap analogs in order to attract the Nature Communications audience. As such, the present revised manuscript is still too thin w.r.t. to the translation outcome but the method has some merit with revised clarity.

[Response_1]

Thank you very much for your time and efforts in reviewing our manuscript. We were pleased to read your comment that the revision we made addressed your queries. According to your suggestion, we have created a table summarizing the capping efficiencies and translation activity data so that the readers can easily understand the outcome of the results. This table has been added to the main manuscript as Table 1 in this revised manuscript. Despite the criticism that there are not enough data here, we believe that what we report in this paper would be of interest to readers of Nature Communications.

We sincerely hope this revision is satisfactory to you.

Hiroshi Abe